# Springtime nitrogen oxides and tropospheric ozone in Svalbard: results from the measurement station network

Alena Dekhtyareva[1,2,3], Mark Hermanson[4], Anna Nikulina[5], Ove Hermansen[6], Tove Svendby[7], Kim Holmén[8,9], and Rune Graversen[9,10]

[1]Geophysical Institute, Faculty of Mathematics and Natural Sciences, University of Bergen, Post box 6050, Bergen, 5020 Norway
[2]Bjerknes Centre for Climate Research, Bergen, Jahnebakken 5, 5007 Norway
[3]Department of Automation and Process Engineering, Faculty of Engineering Science and Technology, UiT The Arctic University of Norway, Post box 6050 Langnes, 9037 Tromsø, Norway
[4]Hermanson and Associates LLC, 200 W 53rd str., Minneapolis, MN 55419, USA
[5]Russian Arctic Scientific Expedition on Spitsbergen, Arctic and Antarctic Research Institute, Beringa str., 38, Sankt-Petersburg, 199397 Russia
[6]Department of Monitoring and Information Technology, NILU - Norwegian Institute for Air Research, Instituttveien 18, Kjeller, 2007 Norway
[7]Department of Atmosphere and Climate, NILU - Norwegian Institute for Air Research, Instituttveien 18, Kjeller, 2007 Norway
[8]International director, Norwegian Polar Institute, Post box 505, Longyearbyen, 9171 Norway
[9]Department of Physics and Technology, Faculty of Science and Technology, UiT The Arctic University of Norway, Post box 6050 Langnes, 9037 Tromsø, Norway
[10]Norwegian Meteorological Institute, Kirkegårdsvegen 60, 9239 Tromsø

**Correspondence:** Alena Dekhtyareva (alena.dekhtyareva@uib.no)

**Abstract.** Svalbard is a remote and scarcely populated Arctic archipelago, and is considered to be mostly influenced by the long-range transported air pollution. However, there are also local emission sources such as coal and diesel power plants, snowmobiles and ships, but their influence on the background concentrations of trace gases have not been thoroughly assessed. This study is based on tropospheric ozone ($O_3$) and nitrogen oxides ($NO_x$) data collected in three main Svalbard settlements in spring 2017. In addition to these ground-based observations, radiosonde and $O_3$ sondes soundings, ERA5 reanalysis and BrO satellite data have been applied in order to distinguish the impact of local and synoptic-scale conditions on the $NO_x$ and $O_3$ chemistry. The measurement campaign was divided into several subperiods based on the prevailing large-scale weather regimes. The local wind direction at the stations depended on the large-scale conditions, but was modified due to complex topography. The $NO_x$ concentration showed weak correlation for the different stations and depended strongly on the wind direction and atmospheric stability. On the contrary, the $O_3$ concentration was highly correlated among the different measurement sites and was controlled by the long-range atmospheric transport to Svalbard. Lagrangian backward trajectories have been used to examine the origin and path of the air masses during the campaign.

# 1 Introduction

Fossil fuel combustion and biomass burning create high-temperature conditions leading to the reaction between atmospheric oxygen and nitrogen present in the fuel and in the air and formation of nitrogen oxides ($NO_x = NO + NO_2$) (Seinfeld and Pandis, 2006). $NO_x$ emitted locally in the Arctic or transported from mid-latitudes may increase the deposition of nitrates ($NO_3^-$), which act as nutrients, and which during the climate change may cause changes in the relative abundances of species in nutrient-deficient environments such as lakes in Svalbard (AMAP, 2006). Aerosols, containing particulate nitrate, are formed from the gaseous nitric acid ($HNO_3$) produced through the oxidation of nitrogen dioxide ($NO_2$) by OH-radicals in the presence of sunlight or by the nighttime reaction with tropospheric ozone ($O_3$) (AMAP, 2006).

High concentrations of $NO_x$ may lead to regional soil and water acidification and have negative effects on human health (AMAP, 2006). In addition to this, $NO_x$ are vital for the formation of tropospheric ozone $O_3$, which is a harmful air pollutant and a greenhouse gas (IPCC Working Group 1 et al., 2013). The $O_3$ production and loss depends on the ratios between hydrocarbons / $NO_x$ and CO / $NO_x$ (carbon monoxide and nitrogen oxides) and the presence or absence of sunlight. In the absence of sunlight during polar night, $O_3$ that has been produced within the long-range transported polluted air masses, may accumulate in the Arctic. Hereby, the atmospheric lifetime of $O_3$ may be increased from days in summer to months in winter (Quinn et al., 2008).

The ultraviolet (UV) solar irradiance has a complex influence on $NO_x$ and $O_3$ chemistry in the troposphere (Seinfeld and Pandis, 2006). Some of the reactions are efficient only at shorter wavelengths, while other processes occur even at longer wavelengths. The insolation increases rapidly in the Arctic during the springtime transition from polar night to midnight sun, but the UV irradiance is dominated by the UV-A fraction with wavelengths from 315 to 400 nm, while the amount of incoming short wave UV-B irradiance with wavelengths from about 300 to 315 nm is still minimal in this period (Seinfeld and Pandis, 2006). One of the processes that takes place even under low solar elevation and higher column ozone concentration over the sea-ice and snow-covered surfaces, is the photolysis of dihalogens (Simpson et al., 2015). This process is the initial step needed for the heterogeneous photochemical reactions with bromine compounds promoting the springtime tropospheric $O_3$ depletion events (Fan and Jacob, 1992; Monks, 2005; Simpson et al., 2015). According to the study performed by Beine et al. (1997a), the background $NO_x$ values were also lower than normal during such events observed at the Zeppelin station in Svalbard. The reactions with Br-species, which result in oxidation of NO to $NO_2$ and removal of $NO_2$ by the reaction with BrO or OH-radical and formation of $HNO_3$, were proposed as possible explanation to this phenomenon. However, lower $NO_x$ values are also characteristic of the pristine air masses from the remote regions in the high Arctic. In contrast, elevated $NO_x$ values are observed during the pollution episodes near the local emission sources or when $NO_x$ are transported to the Arctic from mid-latitudes directly or in the form of peroxyacetylnitrate (PAN), which is further thermally decomposed locally in the Arctic when the air temperature increases in spring (Beine et al., 1997b). Irrespective to the UV irradiance, in the vicinity of large sources of NO, such as cruise ships, the titration of $O_3$ and formation of $NO_2$ may be observed (Eckhardt et al., 2013). However, if NO and CO or hydrocarbons are present in sufficient quantities downwind from the emission source and insolation increases (Wallace and Hobbs, 2006), the photolysis of $O_3$ at the wavelengths below 320 nm may lead to production of OH-

radical in presence of water vapour, which may further yield net $O_3$ production. The UV irradiance affects also $NO/NO_2$ ratio. $NO_2$ dissociates to NO and O in the range of wavelengths from 300 nm to 370 nm. The photodissociation efficiency reduces gradually at higher wavelengths and vanishes at 420 nm (Seinfeld and Pandis, 2006). A diurnal variation in the background

$NO/NO_2$ ratio has been observed in Svalbard in spring, and the increase in this ratio around noon becomes more pronounced from March to May (Beine et al., 1997b). The efficiency of $NO_2$ photolysis and formation of NO and $O_3$ enhances as insolation increases, despite concurrent rapid oxidation of NO by $O_3$ leading to formation of $NO_2$ , a second part of the so-called daytime $NO_x$ null-cycle (Wallace and Hobbs, 2006). Thus, both UV-B and UV-A solar irradiance fractions may have influence on the springtime concentrations of $NO_x$ and $O_3$ in Svalbard and should be taken into consideration.

Meteorological conditions, such as wind speed and direction, air temperature and humidity, affect the formation of aerosols and efficiency of pollution dispersion and deposition. Synoptic-scale north-easterly wind is prevailing in the Svalbard region (Adakudlu et al., 2019), but the mesoscale flow is affected locally by topographical channelling and air density gradient from the inland glaciers to the warmer sea. The most pronounced wind direction is along the valleys or fjords towards the coast (Førland et al., 1997): from south-east in Longyearbyen and Ny-Ålesund and from south-south-east in Barentsburg (Figure 1).

Nevertheless, despite the difference in local wind direction in the settlements, there may be common mesoscale meteorological conditions promoting accumulation of locally emitted pollutants in the atmospheric boundary layer (ABL) at all three sites, such as ABL height variation and atmospheric temperature inversion (Dekhtyareva et al., 2018).

     The main anthropogenic emission sources on the archipelago are related to electricity and heat production: coal power plants in Barentsburg and Longyearbyen and a diesel-fuelled generator in Ny-Ålesund (Dekhtyareva et al., 2016; Vestreng

et al., 2009). The energy demand for heating in Longyearbyen is two times higher in winter than in summer due to the lower temperatures in wintertime. In winter days, the production of energy for heating increases from 06:00 to 09:00 in the morning and then decreases steadily until it reaches its minimum at 03:00 in the night, while in summer the production varies little throughout the day (Tennbakk et al., 2018). In contrast to the energy needed for heating, the energy demand for electricity production is mostly independent on the air temperature. Industry, business and municipal buildings stand for more than 70 %

of the electricity consumption in Longyearbyen. There is a diurnal variation in the power demand with higher daytime values in winter. In summer, the power demand and its diurnal variations are lower, since the mine has reduced operation in July (Tennbakk et al., 2018). The power demand for heating in Ny-Ålesund and Barentsburg varies similarly to Longyearbyen, but the absolute values are different for all three settlements.

     Svalbard residents use cars for transportation within the settlements and snowmobiles for springtime off-road traffic (Vestreng

et al., 2009). There were around 2500 snowmobiles registered at Svalbard in 2007 (MOSJ, 2018), and, according to the report issued by the Norwegian Climate and Pollution Agency (Vestreng et al., 2009), local $NO_x$ emissions from these were three times higher than emissions from the gasoline cars. Current number of snowmobiles is around 2100, and it has been fairly stable since 2011 (MOSJ, 2018).

     In Svalbard, the snowmobile traffic is allowed only in specific zones created to minimize environmental impact from the

usage of motorized vehicles on snow covered and frozen ground (Klima- og miljødepartementet, 2001). Furthermore, because of complex terrain, most of the snowmobile routes are confined to valleys. Therefore, the pollution dispersion is restricted

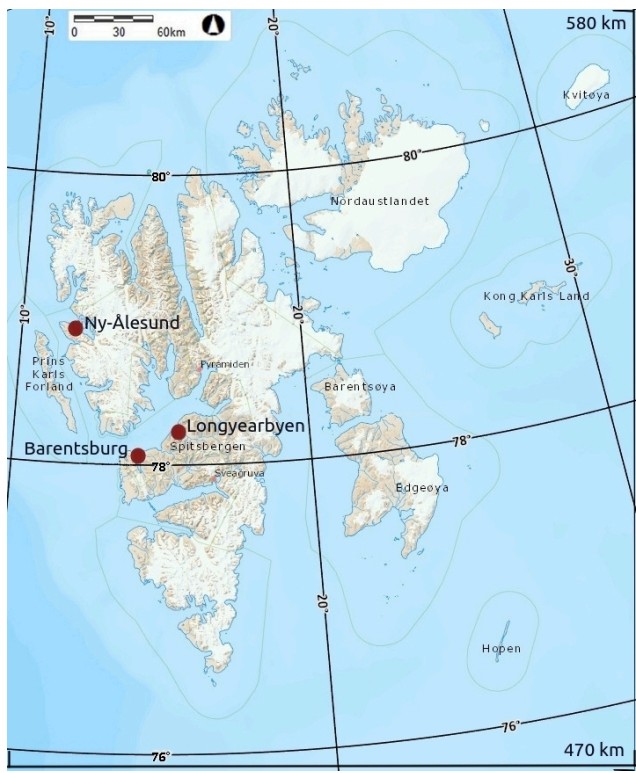

**Figure 1.** Map of Svalbard with three settlements where the $NO_x$ have been measured in spring 2017. The map is made using the online tool https://toposvalbard.npolar.no/ provided by the Norwegian Polar Institute.

and strongly affected by local circulation patterns. The amount of pollutants emitted instantaneously by one motorcade may be significant, since tourists and residents usually travel in groups consisting of up to 20 snowmobiles due to safety reasons. Previous studies have shown highly elevated levels of volatile organic compounds along snowmobile tracks (Reimann et al., 2009), however, no measurements of nitrogen oxides have previously been done.

$NO_x$ concentrations in all three settlements may also be influenced by emissions from the ship traffic (Vestreng et al., 2009; Shears et al., 1998), which is the most intensive in summer (Eckhardt et al., 2013; Dekhtyareva et al., 2016), while snowmobiles and power plants are dominant sources of $NO_x$ in winter and spring seasons.

The main aim of the current article is to combine $NO_x$ and $O_3$ data, collected in spring 2017 in Barentsburg, Longyearbyen and Ny-Ålesund, in order to identify specific factors affecting the concentration of measured compounds and define conditions that promote accumulation of local and long-range transported pollution in all three settlements.

The meteorological *in-situ* and reanalysis data, UV, $O_3$ and $NO_x$ observations have been used to test the following hypotheses:

– There is a diurnal pattern in concentration of $NO_x$ at all three stations due to variable emission rate from the local sources of $NO_x$.

- Complex topography determines local circulation, and therefore variation of $NO_x$ concentration measured at the stations will be dominated by micro- and mesoscale phenomena.

- Local emissions of $NO_x$ in Ny-Ålesund and Barentsburg affect $O_3$ concentrations in the settlements.

- Despite the topographically-induced features, there are common synoptic meteorological conditions, which have an influence on the concentrations of $NO_x$ and $O_3$ in the settlements.

The observational setup at the three stations and methods applied to study various factors of influence on the concentration of measured compounds are introduced in the Materials and Methods (section 2). In the Results (section 3), the $NO_x$ and $O_3$ observations from Adventdalen, Barentsburg and Ny-Ålesund are compared and influence of large-scale weather regimes on the concentrations of measured compounds at the three stations is identified. In the Discussion (section 4), the results from 2017 campaign are contrasted to modelled $NO_2/NO_x$ ratio and $NO_2$ produced through PAN decomposition and long-term observations from Ny-Ålesund, weather regime and trajectory data are utilized to confirm large-scale circulation and air pollution links. Finally, the Conclusion section summarizes main findings of the paper and implications of the weather regime approach to air pollution studies in Svalbard.

## 2 Materials and Methods

### 2.1 Measurements in Adventdalen (Longyearbyen)

In the spring season, the main snowmobile route from Longyearbyen to the east coast of Spitzbergen goes along the road through the Adventdalen valley, and therefore there is daily snowmobile traffic nearby the $CO_2$ laboratory belonging to the University centre in Svalbard (UNIS $CO_2$ lab, coordinates: 78.20247°N 15.82887°E). The station is located at the distance of approximately five km to the south-east from the coal power plant and major cross-roads in Longyearbyen, and thus is representative for monitoring of air pollution from snowmobiles. The chemiluminescence $NO/NO_2/NO_x$ analyser (model T200) was installed at this laboratory for the period from 23.03.2017 to 15.05.2017. The inlet of the sampling hose was secured outside from the window, while the temperature inside the laboratory was kept constant with the help of a thermostat to maintain stable conditions needed for correct functioning of the analyser. The sensor was calibrated weekly using a zero-air generator and a certified NO gas with known concentration (800 ppb), and the hourly average $NO_x$ data were scaled linearly to eliminate zero drift. The automatic weather station (UNIS AWS) is located nearby the UNIS $CO_2$ lab, and the data from that station have been used to assess local meteorological conditions.

In addition to the meteorological parameters from the Adventdalen station, data from UV monitors installed at the UNIS roof in Longyearbyen have been used. The sensors SKU 420 UV-A (315-380 nm) and SKU 430 UV-B (280-315 nm), produced by the SKYE Instruments, were calibrated 24th of August 2016.

## 2.2 Measurements in Ny-Ålesund

Continuous $NO_x$ measurements are performed by the Norwegian institute for Air Research (NILU) in the framework of the air quality monitoring programme in Ny-Ålesund (Johnsrud et al., 2018). The analyser is installed in the middle of the settlement (coordinates: 78.92470°N 11.92641°E), 100 m to the north-west from the meteorological station operated by the Norwegian meteorological institute and 300 m to the south-south-east from the diesel power plant. Similarly to the measurements in Adventdalen, weekly calibrations with zero air and span gas are performed at the Ny-Ålesund station, and the hourly average $NO_x$ data are corrected for drift. The hourly $O_3$ gas monitor data from the Zeppelin observatory, located nearby the mountain top (474m a.s.l.) two km to the southwest from Ny-Ålesund (coordinates: 78.90719°N 11.88606°E), have been used for comparison with the $O_3$ measurements in Barentsburg. The UV data obtained using multifilter radiometer GUV 541 at the Sverdrup station in Ny-Ålesund (Gröbner et al., 2010; Schmalwieser et al., 2017) and local meteorological observations from the Zeppelin station have been provided by NILU as well. The GUV radiometer is checked and corrected against a travelling reference instrument every year.

In addition to this, daily radiosonde and weekly ozone sonde data from the French–German AWIPEV research station in Ny-Ålesund have been used. Since temperature inversion may be an important factor promoting accumulation of local pollution in the atmospheric boundary layer, the method for its detection in the radiosonde vertical profiles, described by Dekhtyareva et al. (2018), has been applied: the days, when the temperature was increasing with height on more than 0.3 °C in the lowest 500 m, were defined as days with temperature inversions. In order to compare the $O_3$ sonde measurements with ground-level observations, the $O_3$ mixing ratio in units of ppbv have been calculated from the $O_3$ partial pressure and atmospheric pressure measured by the radiosonde (Seinfeld and Pandis, 2006).

Daily radiosonde launches are operated at the AWIPEV station (AWI), using Vaisala RS92 radiosondes until April 2017 (Maturilli and Kayser, 2017) and Vaisala RS41 radiosondes afterwards. In this study, we apply radiosonde data for March 2017, post-processed according to the principles of Reference Upper-Air Network GRUAN (Immler et al., 2010). The RS41 data for April-May 2017 are processed with manufacturer's software.

## 2.3 Measurements in Barentsburg

The Russian Arctic and Antarctic Research Institute (AARI) performs the measurements in Barentsburg independently in the frame of the air quality monitoring programme. The equipment installed in the settlement includes Chemiluminescence $NO_x$ AC32M and UV Photometric $O_3$ $O_3$42M Analysers produced by Environnement S.A. and and Vaisala HydroMet system MAWS201. The observational site is located on the narrow terrace 40 m above sea level (coordinates: 78.06070°N 14.21718°E) and 600 m to the north-east from the coal-power plant. The analysers continuously gather the data and transmit them to the laboratory facility of the Russian Scientific Centre in Barentsburg. The data with 20-minutes time resolution have been averaged to obtain hourly data. The analysers have been installed and initially calibrated by the manufacturer's accredited specialists in December 2016. After the installation, the zero control was performed regularly using the $NO_x$ analyser's built-in automatic zero air function for the correction of zero drift lines. The maintenance of the converter filter has been done at the frequency

recommended by the manufacturer. However, in contrast to the equipment in Ny-Ålesund and Longyearbyen, the $NO_x$ and $O_3$ analysers in Barentsburg have not been calibrated manually during the field campaign. Therefore, the data from this station may be prone to drift. This is especially important to keep in mind when studying $NO_x$ concentrations, since the $NO_x$ values are usually very low in the remote Arctic environment (Dekhtyareva et al., 2016). On the other hand, the UV $O_3$ monitor is more stable and does not demand as frequent calibration as chemiluminescence instruments (Williams et al., 2006), and thus data from this instrument are more reliable.

### 2.4 Methods to study the effect of meteorological conditions on the concentration of measured compounds

Previous studies showed influence of large-scale weather phenomenon on long-range transported and local air pollution. Eckhardt et al. (2003) demonstrated how positive and negative phases of North-Atlantic oscillation control long-range transport of air pollution to the Arctic. Modelling study of Ménégoz et al. (2010) introduced four weather regimes based on ERA40 data and described the aerosol budget variations associated with different regimes and feedback of aerosol distribution on the weather regime persistence. Ibrahim et al. (2021) classified nine weather regimes using self-organizing maps and cluster analysis of principal components and investigated the influence of the prevailing large-scale meteorological conditions on the air quality in Berlin. In the current work, we apply Dr. Christian Grams's weather regime classification that is based on the 6-hourly ERA-Interim data. This classification was previously used to investigate frequency of poleward moisture transport events by atmospheric rivers (Pasquier et al., 2019), southward transport of Arctic air during cold-air outbreaks (Papritz and Grams, 2018) and for assessment of the Europe's wind power output (Grams et al., 2017). Thus, this approach is suitable for study of long-range transport of air masses and local dispersion efficiency that depends on the atmospheric stability and wind speed. The following three-step procedure has been implemented to assess the effect of the prevailing synoptic meteorological situation on long-range transport of pollutants and on the local meteorological conditions affecting concentrations of $O_3$ and $NO_x$ in Svalbard.

Firstly, the whole measurement period has been divided into sub-periods based on the prevailing large-scale atmospheric circulation patterns. In the climatological mean, the large-scale conditions are characterized by weak ridging of absolute geopotential height at 500 hPa over the eastern North Atlantic and westerly upper level flow over Svalbard. Such regime is placed in the "no regime" category in the Grams et al. (2017) classification (their Fig. S1h). The deviations from these mean conditions are classified into seven distinct weather regimes that represent the variation of the large-scale circulation patterns over the North Atlantic and European region.

Secondly, the hourly meteorological data, $O_3$ concentration and planetary boundary layer height (PBL) from the global ERA5 reanalysis data set with 31 km spatial resolution (Hersbach et al., 2020) have been used to investigate the prevailing large-scale weather conditions for the identified sub-periods. The ERA5 $O_3$ mass mixing ratio is estimated based on the assimilated satellite observations and external 2-D photochemical model (Park et al., 2020). The PBL in the ERA5 data set is defined by the height when the Richardson number for the adjacent vertical model layers exceeds a critical value of 0.25, and the air becomes stably stratified (European Centre for Medium-Range Weather Forecasts, 2017).

Thirdly, the FLEXible PARTicle (FLEXPART) V8.2 air parcel trajectory data sethas been utilized for the same sub-periods to study the long-range atmospheric transport to Svalbard. Previously, FLEXPART data have been used to investigate long-range transport of black carbon and sulphates (Hirdman et al., 2010a, b), mercury and $O_3$ (Hirdman et al., 2009) in the Arctic. The current FLEXPART data set contains a 3-dimensional Lagrangian dispersion simulation results (Stohl et al., 2005) with

5 million air parcels (Läderach and Sodemann, 2016; Fremme and Sodemann, 2019)driven with the meteorological data from the ERA-Interim reanalysis with a spatial resolution of approximately 80 km and temporal resolution of 6 hours (Dee et al., 2011). The 10-days backward trajectories starting within the lowest 500 m above the ground in the region covering both Ny-Ålesund, Longyearbyen and Barentsburg (from 77.5°N to 79.5°N and from 10°E to 20°E) have been extracted from this data set. The trajectory model data have been combined with BrO total column data derived from GOME-2 (ir)radiance satellite

observations (AC SAF, 2017) to identify regions with elevated concentration of this $O_3$ depleting substance.

Additionally, to study long-range transport of extremely $O_3$-depleted or enriched air masses, the following procedure has been implemented to detect $O_3$ decrease and increase events occurring simultaneously in Barentsburg and at the Zeppelin station:

1. Since the distance between the Zeppelin observatory and Barentsburg is more than 100 km, a time lag in correlation

between the data from the two stations is expected. The acceptable time lag has been defined based on the lagged linear correlation between the datasets. Maximum time lag, for which the correlation coefficient is higher or equal to the coefficient calculated for the zero-hour lag, is defined as maximum allowable time lag.

2. Applying the air-quality extreme definition stated in Porter et al. (2015), $O_3$ levels below the 5th quantile and above the 95th quantile have been found separately for the Barentsburg and Zeppelin measurements to define severe depletion and

increase events, respectively.

3. Continuous episodes have been defined for the periods where the time difference between consecutive event points is less than 3 hours.

4. Minimum (maximum) $O_3$ concentrations within each event have been defined. The time of minimum (maximum) within the events in Barentsburg and at the Zeppelin station have been compared and if the difference between them is less than

the acceptable time lag, the events at both stations have been classified as one joint event.

The backward air mass ensemble trajectories have been simulated using the Hybrid Single Particle Lagrangian Integrated Trajectory (HYSPLIT) model for these joint events for 240 hours back in time to identify the source regions of the air masses (Stein et al., 2015). This 10-days simulation time has been chosen as a compromise between the average lifetime of tropospheric $O_3$, which may be three to four weeks (Christiansen et al., 2017), and the uncertainty of modelled air mass trajectories that

increases with travelled distance (Freud et al., 2017). The standard ensemble simulation with 27 members was calculated in the READY system by offsetting the Global Data Assimilation System (GDAS) meteorological data with a $0.5^o$ resolution by a fixed grid factor in the horizontal and vertical dimensions to take into account the uncertainty of the trajectory forecast (Rolph et al., 2017).

## 2.5 Methods to study the effect of local pollution in Ny-Ålesund and in Barentsburg on the measured $O_3$ concentrations

Previously, the only collocated $NO_x$ and $O_3$ measurements from Ny-Ålesund were performed at the Zeppelin station from February to May 1994. The results were published in Beine et al. (1996). In that study, the combination of $NO_x$ data and concentration of particles with diameter below 10 nm, atmospheric stability and wind direction was used to identify possible local pollution events. In spring 1994, the local pollution was detected at the Zeppelin station during 6.4 % of the measurement time, and the number of these events was increasing with increased insolation in May. As there were no $O_3$ data from Ny-Ålesund available, the $O_3$, CO and The Differential Mobility Particle Sizer (DMPS) data from the Zeppelin station were used to study the influence of local $NO_x$ emissions in Ny-Ålesund on the $O_3$ concentration. CO indicates presence of other pollutants emitted simultaneously in the process of fossil fuel burning, and although the correlation between $NO_x$ and CO concentration in the plumes depends on the engine and fuel type, age of the plume and environmental conditions (Li et al., 2015), we expect elevated CO concentrations in the fresh plumes arriving to the Zeppelin station. Therefore, a local pollution effect has been defined for $O_3$ measurements at the Zeppelin station when all four conditions were fulfilled:

1. the wind direction measured both in Ny-Ålesund and at the Zeppelin station was northerly (above $270^o$ or below $90^o$) since the diesel power plant is located in 300 m to the north-north-west from the $NO_x$ monitor in Ny-Ålesund and 2 km to the north-north-east from the Zeppelin station;

2. $NO_x$ concentrations were above the mean value in Ny-Ålesund;

3. the concentration of particles with diameter below 10 nm had a strong increase (above 95 percentile value for the whole campaign);

4. CO concentrations observed at the Zeppelin station were above the mean value indicating the possible impact of local pollution.

To assess how the $NO_x$ emissions in Barentsburg affect the local $O_3$ concentration there, the $NO_x$ and $O_3$ data from the Barentsburg station have been compared. Positive anomalies in $O_3$ concentration were found for the same wind directions where increased $NO_x$ concentrations were observed, but this may be due to higher concentrations of $O_3$ in air masses, which were transported to Svalbard from the south and south-west. Since there are multiple sources of local pollution in Barentsburg (coal power plant, ships and cars), another method has been implemented:

1. the hours when $NO_x$ concentrations were above average in Barentsburg have been defined;

2. $O_3$ values for these hours in the original and in the 6-hourly smoothed data series from Barentsburg have been compared.

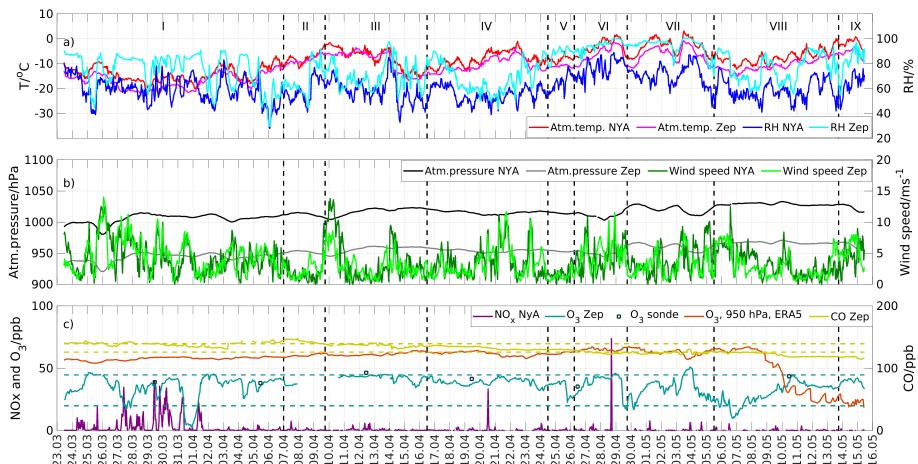

**Figure 2.** (**a**) The hourly average air temperature and relative humidity measured in Ny-Ålesund (NYA) and at the Zeppelin station (Zep); (**b**) Atmospheric pressure and wind speed measured in Ny-Ålesund and at the Zeppelin station; (**c**) $NO_x$, CO and $O_3$ concentrations in Ny-Ålesund, at the Zeppelin station and in the ERA5 reanalysis for the nearest grid point the coordinates of Ny-Ålesund for the period from 23th of March 2017 to 16th of May 2017. The dashed black lines represent time frames of the different weather regimes. The dashed blue line shows 5th and 95th quantiles of the $O_3$ concentration.

## 3 Results

### 3.1 Comparison of $NO_x$ and $O_3$ observations from Adventdalen, Barentsburg and Ny-Ålesund

The hourly $NO_x$ and $O_3$ concentrations and meteorological data from all three stations are shown in Figure 2 and Figure 3.

Nine distinct large-scale weather regimes have been identified for the campaign period (marked with numbers from I to IX). The detailed description of the weather regimes is given the part 3.2 of the current paper.

Despite alteration in the large-scale circulation, little variability was found in CO data from the Zeppelin station (solid olive line in the Figure 2 c)) and $O_3$ values in ERA5 reanalysis (orange line n the Figures 2c)). The threshold of CO median value $\pm$ median absolute deviation (132.6 ppb $\pm$ 6.7 ppb) is shown by the dashed olive line in the Figure 2c). There were no sharp

peaks in the concentration of this gas indicating that there was little influence of local pollution on the measurements at the mountain station, but the levels of this compound were stably high in the beginning of the campaign and showed gradual decline from the end of March to the middle of May. This is a response to increased insolation throughout the field work period (from 0.03 $W \cdot m^{-2}$ to 0.3 $W \cdot m^{-2}$ and from 7.9 $W \cdot m^{-2}$ to 28.9 $W \cdot m^{-2}$ UV-B and UV-A irradiation observed in Ny-Ålesund, respectively), as CO is rapidly oxidized by the OH-radical produced in the $O_3$ photolysis. A pronounced decline in

CO concentration and sharp drop in $O_3$ values in ERA5 reanalysis may be observed in sub-period VIII. The $O_3$ concentration in the ERA5 reanalysis exceeds the values observed at the Zeppelin station (dark blue line in the Figure 2c) for most of the time except this sub-period. Thus, the $O_3$ ERA5 reanalysis data is not representative for the regional Arctic processes or short-term variability in long-range transport, but is showing strong sensitivity to photochemical destruction.

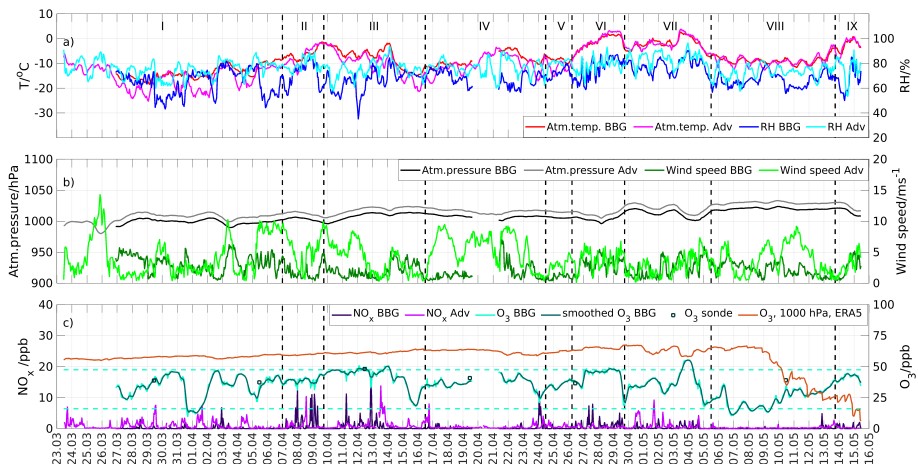

**Figure 3.** (**a**) The hourly average air temperature and relative humidity measured in Adventdalen (Adv) and Barentsburg (BBG); (**b**) atmospheric pressure and wind speed measured in Adventdalen and Barentsburg; (**c**) $NO_x$ and $O_3$ concentrations in Adventdalen, in Barentsburg and in the ERA5 reanalysis for the nearest grid point to the coordinates of Barentsburg for the period from 23th of March 2017 to 16th of May 2017. The dashed black lines represent time frames of the different weather regimes. The dashed light blue line shows 5th and 95th quantiles of the $O_3$ concentration.

There is a weak statistically significant positive correlation between NO, $NO_2$ and $NO_x$ values measured in Adventdalen and in Ny-Ålesund (the Pearson correlation coefficients are rNO=0.13, $rNO_2$=0.15 and $rNO_x$=0.13, p<0.0001 for all compounds). On the contrary, no correlation is present with $NO_x$ data from Barentsburg. Low correlation between the $NO_x$ values at the three stations indicates the importance of local emission sources and micrometeorology (wind channelling and spatial variation in atmospheric stability) rather than synoptic meteorological conditions. The background $NO_x$ concentrations observed in Svalbard in previous studies (Beine et al., 1997a, b) using different measurement techniques are below 0.4 ppb, and thus, the natural variability in $NO_x$ values due to long-range transport to Svalbard would be undetected in the $NO_x$ datasets presented in the current study.

The Barentsburg $O_3$ data contains some abrupt peaks with magnitude of up to 9 ppbv and duration of just one hour (light blue line in Figure 3c)), while they are absent in the Zeppelin $O_3$ data. Indeed, the Barentsburg station is located inside the settlement, and the $O_3$ data there is more prone to be influenced by the local $NO_x$ pollution, while the Zeppelin station is situated far from the local emission sources. In order to investigate the significance of this local impact, a 6-hour moving average filter has been applied to the $O_3$ Barentsburg data, and the results are shown with a dark blue line in Figure 3c). The smoothed and original $O_3$ data from Barentsburg have been compared statistically: both two-sided Wilcoxon rank sum (WRS) test and the t-test show that the application of the low-pass filter on the $O_3$ data from Barentsburg does not result in significant change in the concentration distribution. The correlation between $O_3$ concentrations at the Zeppelin station and in Barentsburg is moderate (Pearson correlation coefficient r=0.69 both for smoothed and unsmoothed data, p<0.001). This indicates that $O_3$

concentrations at both stations are highly influenced by the meteorological conditions on the synoptic scale and local impacts are of minor importance.

We have applied methods described in section 2.5, to define the effect of local $NO_x$ emissions in Ny-Ålesund and Barentsburg on the $O_3$ concentrations in the settlements. As a result, 5% of the $O_3$ data from the Zeppelin station might have been
influenced by the local pollution from Ny-Ålesund, and the statistically significant (p<0.0001) decrease in the $O_3$ mean (31.6 vs 36.1 ppb) and median (34.4 vs 38.0 ppb) concentrations have been revealed for this group. However, northerly wind that may transport local pollution from Ny-Ålesund also brings air masses, which have lower $O_3$ background value. Therefore, when one compares potentially locally polluted air masses with the background air masses coming from the north, the difference in mean and median $O_3$ concentrations becomes statistically insignificant, 31.6 vs 33.3 ppb and 35.6 vs 34.4 ppb, respectively.
Following the method of Beine et al. (1996) for the local pollution event detection, the concentration of particles with diameter of 10 nm routinely measured by DMPS at the Zeppelin station and threshold of > 95 percentile have been used to identify peaks in concentration of newly formed particles. Similarly to the results of Beine et al. (1996), the peak events were detected at the Zeppelin station only in the second part of 2017 measurement campaign (from 24th of April to 13th of May). The northerly wind direction was present only during 12 out of 45 hours with peak particle concentration, however, none of these cases
was characterized by increased CO concentration at the Zeppelin station. Thus, these peaks in concentration of small particles might have been related to natural rather than anthropogenic emission sources. Indeed, Heintzenberg et al. (2017) described the offset in new particle formation towards late spring and summer when biological emissions become important sources for this process. Therefore, both statistical comparison of the $O_3$ concentrations in clean and potentially polluted air masses mentioned above and absence of coinciding peaks in particle concentration and CO concentration indicate that the $O_3$ observations at the
Zeppelin station were not significantly affected by local $NO_x$ pollution during 2017 campaign.

Difference between the original and smoothed $O_3$ data from Barentsburg varies from -19% to 11% of the smoothed value, and there is a moderate negative correlation between the magnitude of $NO_x$ peak and the reduction in $O_3$ concentration (r=-0.65, p<0.0001). Despite this sensitivity of $O_3$ concentration to local pollution in Barentsburg, the median $NO_x$ concentrations observed there were low and average reduction of $O_3$ concentration in comparison to the smoothed values was only 1%.
This effect is not statistically significant, and therefore other factors, such as variation in concentrations within long-range transported air masses, may be more important for explanation of difference between the $O_3$ Zeppelin and Barentsburg datasets.

The comparison of the vertical $O_3$ data from the $O_3$ sondes (dark blue squares in Figure 2c) and 3c)) from Ny-Ålesund and the ground-based measurements at the Zeppelin station and in Barentsburg reveals that the discrepancy in the data between the two stations may be explained by the fact that the stations are located at different heights and measure air masses with uneven distribution of $O_3$ in the lowest 500 m. If contrasting the closest point to the sounding time in the observations made in
Barentsburg and in Ny-Ålesund, similar tendencies as in the $O_3$ sonde data may be observed. For example, there is a significant difference between the data from Barentsburg and Ny-Ålesund (33.74 ppb vs 38.88 ppb) for the measurement closest to the time of sounding on 10th of May, and increase of $O_3$ concentration with height between 1000 hPa (50 m) and 950 hPa (500 m) is noticeable in the sounding data as well (Figure 4a). One can see in the potential temperature profiles (Figure 4b) that the
pronounced atmospheric inversion tend to be noticeable in the $O_3$ sonde profiles as well. For example, simultaneous increase

in $O_3$ concentrations and potential temperature is pronounced at the level of 850 hPa (1300 m) on 26th of April, 860 hPa (at 1200 m) on 5th of April and 900 hPa (at 1000 m) on 12th of April (Figures 4a) and 4b)). The $O_3$ concentration in the ERA5 profiles was overestimated and showed little variability in the lowermost layer, except the profile for 10th of May (dashed dark gray line in the Figure 4a)) when the reanalysis and observational values coincided for 1000 hPa level, but the $O_3$ concentration in ERA5 was underestimated by almost 40% at the height of 925 hPa (Figures 4a). The virtual potential temperature profiles in reanalysis resemble $O_3$ sonde profiles closely (Figure 4b). The PBL height from ERA5 reanalysis (marked with stars in the Figure 4b)) was above the low-level temperature inversions detected on 5th and 12 of April and below the level of the most pronounced virtual temperature inversion in the $O_3$ sonde profiles.

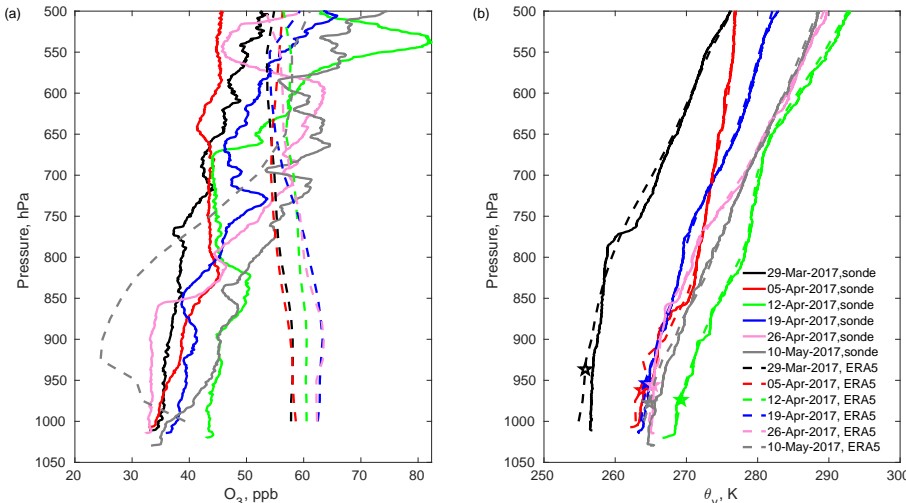

**Figure 4.** Vertical profiles of $O_3$ (a) and virtual potential temperature (b) from $O_3$ sondes and ERA5 reanalysis data. $O_3$ sonde profiles are shown with solid lines and ERA5 profiles are shown with dashed lines. The planetary boundary layer height (PBL) in the ERA5 data is plotted with stars on the virtual potential temperature profiles from the reanalysis.

In addition to hourly values, average concentrations of measured compounds have been calculated for each hour of the day. The diurnal variation in NO, $NO_2$ and $O_3$ concentrations at the stations is shown in Figure 5. Mean and median daytime (from 6:00 UTC to 17:00 UTC) and nighttime (from 18:00 UTC to 5:00 UTC) concentrations are shown in Table 1 (here the daytime and nighttime are defined based on snowmobile traffic pattern in the Adventdalen valley).

The $NO_2/$ $(NO+NO_2)$-ratio is quite high in Adventdalen and in Barentsburg and exhibits diurnal variation, while it is much lower in Ny-Ålesund and there is no statistically significant difference between its day and night values. This may be explained by the fact that the measurement station in Ny-Ålesund was located much closer to the diesel power plant, a constant source of fresh $NO_x$ emissions, where the $NO_2/NO_x$ ratio is much lower irrespective to the time of the day (Beine et al., 1996). The lowest hourly $NO_2/NO_x$ ratio of 0.29 and the highest peak of $NO_x$ were observed in Ny-Ålesund at 17:00 UTC 28th of April (Figure 2). The concentration of NO and $NO_2$ were 87.8 ppb and 16.4 ppb, respectively, which indicates the presence of a strong emission source, for example snowmobiles, in the immediate vicinity from the station. Since this was a single $NO_x$ peak in the data, $NO_2/NO_x$ ratio was unusually low and the meteorological conditions were untypical for pollution accumulation in Ny-Ålesund (south-easterly wind with moderate speed of 5.3 $ms^{-1}$), hence this value has been excluded from further statistical analysis.

NO is a primary product of fossil fuel combustion (Seinfeld and Pandis, 2006; Arya, 1999), and higher $NO/NO_x$ ratio is expected close to the emission source. The station in Adventdalen is located at a distance of five kilometres from the coal power plant, and snowmobile traffic there is a temporarily emission source present mostly during daytime. In contrast, measurement stations in Barentsburg and Ny-Ålesund are located near the power plants constantly releasing combustion products at a variable rate. Thus, it is noticeable in Adventdalen that the NO concentration is close to zero during the night (dark blue bar in Figure

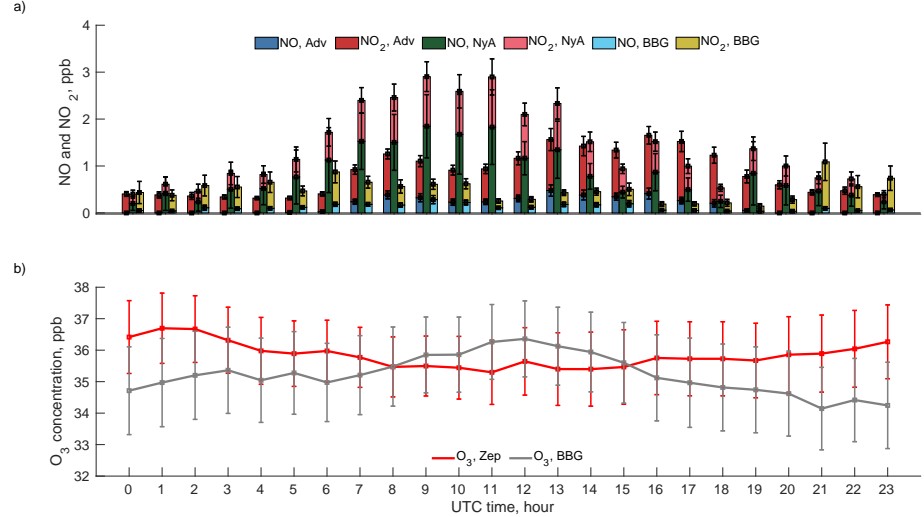

**Figure 5.** (**a**) Variation of measured NO and $NO_2$ concentrations depending on the time of the day (in UTC) in Adventdalen (Adv), Ny-Ålesund (NyA) and Barentsburg stations (BBG). (**b**) Variation of measured $O_3$ concentration depending on the time of the day (in UTC) at the Zeppelin (Zep) and Barentsburg stations (BBG). The whiskers show standard error of the mean for each group.

5a)) in absence of fresh traffic emissions and photochemical conversion of $NO_2$ to NO. As the traffic intensity increases during the day, the NO concentration rises, however, so does the $NO_2$ concentration (red bar in Figure 5a)) since there is a rapid
conversion of NO to $NO_2$ by the reaction with $O_3$.

There is a slight increase in the $O_3$ values in Barentsburg (gray line in Figure 5b)) during daytime. In contrast, a slight decrease in daytime $O_3$ concentration is observed at the Zeppelin station. The discrepancy in the diurnal $O_3$ profiles from the two stations may be explained by the difference of the altitude of each of the two stations and response of the measurements to the boundary layer dynamics. The ERA5 data show diurnal variation in the PBL height for both locations with highest average
values of 462 m and 293 m, in Barentsburg and Zeppelin, respectively, for 13 UTC. The Barentsburg station is located at the altitude of 50 m a.s.l. and samples air within the ABL. During the daytime, the vertical mixing between the atmospheric surface layer and the air masses aloft enhances, and the boundary layer height increases. This mixing process may enrich the surface layer with $O_3$. Similar influence of convection on replenishing the $O_3$ in the Arctic ABL after the depletion events has been described in the work of Moore et al. (2014). In contrast, the Zeppelin station is located at the altitude of 474 m a.s.l.
and mostly samples air from the free troposphere with higher $O_3$ concentration, and thus the data from this station does not exhibit similar diurnal variation as the Barentsburg station. However, the magnitude of these effects is small, and according to the t-test and the WRS-test, there is no statistically significant difference between the nighttime and daytime $O_3$ concentrations measured at the stations (Table 1).

The average NO and $NO_2$ concentrations measured at the stations are distributed unevenly over the wind directions. In
Adventdalen, the average wind speed was $5.1 \pm 3.1\ ms^{-1}$. The south-easterly wind was dominating during the field campaign (Table 2), and there was no significant difference between the daytime and nighttime observations. The highest average daytime

**Table 1.** Measurement results from Adventdalen, Barentsburg and Ny-Ålesund. The two-sided t-test compares daytime and nighttime concentrations at each station and checks if there is a significant difference in mean values for these two groups. The two-sided WRS-test compares daytime and nighttime concentrations at each station and checks if there is a significant difference in median values for these two groups. Pairs with significant (p<0.05) t- and WRS-test results are shown with bold font.

| Compound and station | Daytime mean value | Nighttime mean value | p-value, t-test | Daytime median value | Nighttime median value | p-value, WRS-test |
|---|---|---|---|---|---|---|
| NO, ppb: | | | | | | |
| Adventdalen | **0.30** | **0.00** | 0.000 | **0.12** | **0.00** | 0.000 |
| Barentsburg | **0.15** | **0.08** | 0.000 | **0.03** | **0.01** | 0.000 |
| Ny-Ålesund | **1.29** | **0.52** | 0.001 | **0.14** | **0.02** | 0.000 |
| | | | | | | |
| $NO_2$, ppb: | | | | | | |
| Adventdalen | **0.94** | **0.41** | 0.000 | **0.49** | **0.28** | 0.000 |
| Barentsburg | 0.39 | 0.54 | 0.068 | 0.00 | 0.00 | 0.099 |
| Ny-Ålesund | **0.82** | **0.33** | 0.000 | **0.15** | **0.03** | 0.000 |
| | | | | | | |
| $NO_2/(NO+NO_2)$ ratio: | | | | | | |
| Adventdalen | **0.80** | **0.83** | 0.009 | **0.82** | **0.85** | 0.000 |
| Barentsburg | **0.72** | **0.80** | 0.000 | **0.78** | **0.89** | 0.000 |
| Ny-Ålesund | 0.61 | 0.63 | 0.369 | 0.64 | 0.63 | 0.335 |
| | | | | | | |
| $O_3$, ppb: | | | | | | |
| Barentsburg | 35.64 | 34.81 | 0.139 | 37.33 | 35.74 | 0.051 |
| Zeppelin | 35.55 | 36.14 | 0.203 | 37.18 | 38.28 | 0.057 |

NO and $NO_2$ concentrations were observed when the wind was from north-east and south-east in Adventdalen (Figure 6). To test if the size of snowmobile motorcade has an influence on the $NO_x$ concentration in Adventdalen, manual observations of number of snowmobiles were done in 19 days. In general, the effect of large number of snowmobiles was noticeable in the
$NO_x$ data only in case of low wind speed. For example, in the evening of 01.05.2017, the wind speed was 1.9 $ms^{-1}$, and the $NO_2$ concentration increased sharply to 7.3 ppb due to 21 snowmobiles passing by the station. The group of similar size was passing by in the evening of 02.05.2017, but the effect on $NO_2$ values was three times lower as the wind speed was higher (4.0 $ms^{-1}$). The maximum hourly $NO_2$ concentration of 11.4 ppb was measured on Easter holiday, 13.04.2017. In that day, the combination of increased recreational traffic and mild weather conditions (wind speed below 1 $ms^{-1}$ and air temperature -8°C)
led to accumulation of concentration 13 times higher than daytime hourly average measured during the field campaign. Such low wind speed is untypical for the wind regime in Adventdalen, where normally ventilation is sufficient to effectively disperse

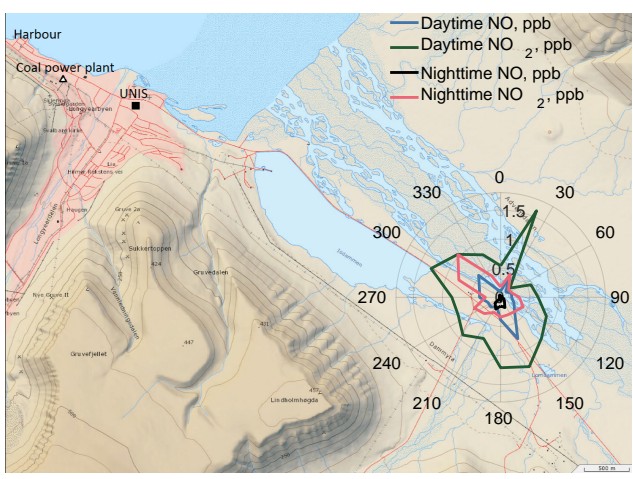

**Figure 6.** Distribution of average NO and $NO_2$ concentrations over wind directions in daytime and nighttime at the station in Adventdalen. The background map is made using the online tool https://toposvalbard.npolar.no/ provided by the Norwegian Polar Institute.

$NO_x$ emitted by the usual amount of motorized traffic. The highest average nighttime $NO_2$ concentrations were detected when the wind was from north-west, which reveals possible influence of the coal power plant. The average nighttime concentrations of NO were very low regardless of the wind direction.

Figures 7a) and 7b) illustrate distribution of NO and $NO_2$ concentrations over wind directions in Ny-Ålesund and Barentsburg, respectively. South-easterly wind with average speed of 3.6 $ms^{-1}$ and 3.9 $ms^{-1}$ in daytime and nighttime, respectively, was dominating in Ny-Ålesund. However, the highest average $NO_x$ concentrations in Ny-Ålesund were measured when the wind was coming from the north (Figure 7a)). This points clearly to the local diesel power plant being the main emission source. Similar results regarding the influence of the local power plant in Ny-Ålesund on $NO_x$ concentrations were presented

in Dekhtyareva et al. (2016) and Johnsrud et al. (2018). During the field campaign, the prevailing wind in Barentsburg was from south and south-east with average speed of 2.5 $ms^{-1}$ and from south-east and east with mean speed of 2.3 $ms^{-1}$ in daytime and nighttime, respectively. The $NO_x$ concentrations measured there were much lower and more evenly distributed over different wind directions than in Ny-Ålesund (Figure 7b)). The coal power plant operates day and night and, in the light wind conditions, may contribute to accumulation of local pollution in the settlement even in absence of south-westerly wind.

**3.2    Influence of large-scale weather regimes on the concentrations of measured compounds at the three stations**

The overview of median concentrations of measured compounds and prevailing local meteorological conditions for the nine sub-periods defined based on the prevailing weather regimes is given in Table 2. The wind directions observed at the stations during each of the sub-periods have been sorted into 16 bins with $22.5^o$ interval. The three main wind directions for each sub-period and for the whole campaign are stated with letters in the Table 2. The detailed analysis of the meteorological conditions

prevailing during each of the weather regimes and their influence on the concentration of the compounds measured at the three station is given below.

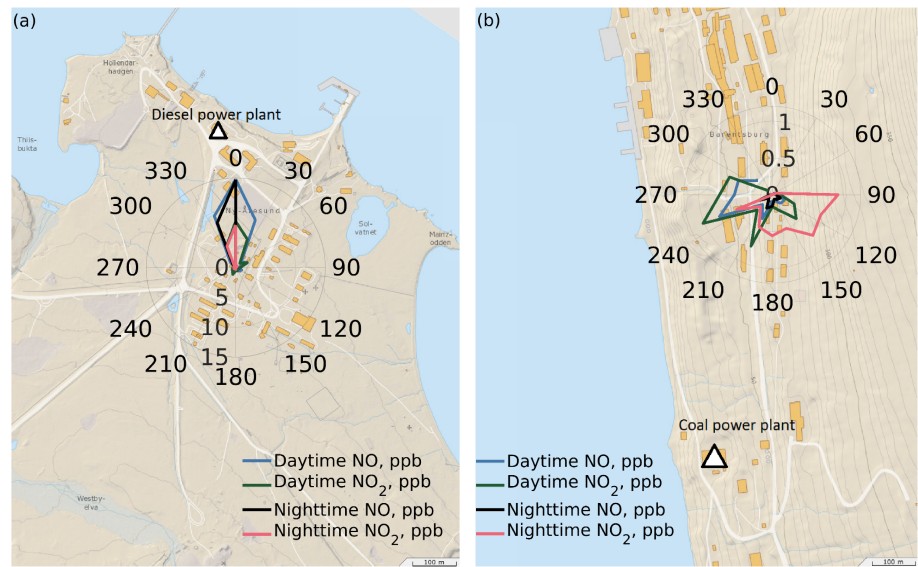

**Figure 7.** Distribution of average NO and $NO_2$ concentrations over wind directions in daytime and nighttime at the stations in Ny-Ålesund (a) and Barentsburg (b). The background maps are made using the online tool https://toposvalbard.npolar.no/ provided by the Norwegian Polar Institute.

The "no regime" conditions were present for almost 20 days or 37% of total campaign duration. The sub-period I lasted for 14 days and was characterized by the lowest median temperatures and UV irradiance at all stations and by synoptic scale north-easterly flow (Figure 8a)). However, the conditions were inhomogenious for this period: there were two quick passing cyclones on the 26th of March and 03rd of April (pressure drops in Figures 2b) and 3b)) that led to increase in local air temperatures and wind speeds. In the sub-period I, the temperature inversions were observed in 46% of the radiosonde profiles from Ny-Ålesund, but the median inversion strength was below $0.95^oC$ (median for the whole campaign).

The sub-periods II-V were characterized by Atlantic ridge (AR) and Scandinavian trough (ScTr). These regimes are characterized by the varying degree of geopotential height ridging over the North Atlantic at 500 hPa. During the AR regime, an upper-level north-westerly flow prevails (Grams et al., 2017). The strongest wind speed was observed in Adventdalen during the AR regimes, since the synoptic-scale lower-level flow (Figure 8b) and d)) was parallel to the Adventdalen valley (Figure 6). During the transition from AR to ScTr, the upper-level ridge weakened and shifted southwards (Grams et al., 2017) as did the cyclonic systems on the lower level (Figure 8c) and e)). The AR and ScTr regimes were characterized by the PBL height being below the median value for the campaign at all sites. The temperature inversions were observed for the regimes II, III and IV with the inversion frequency of 67%, 57%, 13%, respectively, and the inversion strength above the median for the campaign.

The sub-period VI was a three-days "no regime" transition between the AR and ScTr and two blocking regimes: Scandinavian blocking (ScBL) and Greenland blocking (GL). This sub-period was characterized by the synoptic scale westerly wind bringing warm Atlantic air over (Figure 8f)), increasing local temperature and PBL height and adding the westerly component to the wind direction at all stations.

During the sub-period VII (ScBL), the positive geopotential height anomaly was located over northern Scandinavia, and part of the upper-level flow was deflected poleward around the blocking anticyclone (Grams et al., 2017). The anticyclonic movement was pronounced in the lower level flow (Figure 8g)), and the synoptic scale north-westerly flow prevailed over the western part of Svalbard. The local wind speed, temperature and PBL height decreased.

The sub-period VIII (GL) was characterized by the strong positive anomaly in the geopotential height at 500hPa over Greenland and the prevailing upper level north-westerly wind (Grams et al., 2017). The lower-level blocking over Greenland promoted north-easterly flow over Svalbard (Figure 8h)).

The sub-period IX ("no regime") was characterized by the strong anticyclone over the Barents sea, that led to pronounced transport of warm Atlantic air with southerly flow to Svalbard (Figure 8h)). No temperature inversions were detected in the radiosonde data from Ny-Ålesund for the sub-periods VII, VIII and IX.

The elevated $NO_x$ concentrations were observed in Ny-Ålesund and Adventdalen during the sub-periods I-V. This may be explained by the enhanced accumulation of locally emitted $NO_x$ in the ABL due to suppressed vertical mixing in cold days associated with the AR and ScTr regimes. In Adventdalen, $NO_x$ concentrations are not dependent on the wind direction (Figure 6 and Table 2), and east-south-easterly and south-easterly are the dominant wind directions for all regimes except for the period VI. The highest median values of $NO_x$ were observed during the periods with the lowest PBL height, ScTr regimes. During the periods VI and VII, the PBL height increased, but westerly and west-north-westerly wind might have brought pollution from the coal power plant and Longyearbyen town to the Adventdalen valley (Figure 6). In Ny-Ålesund, the highest median $NO_x$ value was observed in sub-period I due to the presence of north-north-westerly wind that brought plume from the power plant to the measurement station (Figure 7a)).

In contrast to Adventdalen, the boundary layer height and cold temperatures played a secondary role for the $NO_x$ concentrations in Barentsburg, and the controlling factor was south-westerly component of the wind for the most polluted periods. The highest median $NO_x$ values were detected during the sub-period VI, when south-south-westerly wind was dominating. The major emission sources in Barentsburg are located on the seashore, and warmer marine air from west and south-west may bring local pollution to the station situated on the hill above these sources (Figure 7b). However, the second highest median $NO_x$ value was observed for the sub-period II, the period with the lowest PBL height and easterly wind at this station. The wind direction was not from the coal power plant, but the wind speed was very low, and thus, the local pollution could accumulate in the ABL if a strong inversion was present aloft.

**Table 2.** Median values of measured parameters and the three most often observed wind directions for different weather regimes and for the whole campaign. The concentrations exceeding median value for the whole campaign are shown with bold font.

| Parameter | 23/03-07/04: no | 07/04-09/04: AR | 09/04-16/04: ScTr | 16/04-24/04: AR | 24/04-26/04: ScTr | 26/04-29/04: no | 29/04-05/05: ScBL | 05/05-13/05: GL | 13/05-15/05: no | Whole Campaign |
|---|---|---|---|---|---|---|---|---|---|---|
| *$NO_x$, ppb:* | | | | | | | | | | |
| Adventdalen | **0.44** | **0.42** | **0.54** | 0.31 | **0.55** | **0.36** | **0.36** | 0.22 | 0.26 | 0.35 |
| Barentsburg | 0.01 | **0.38** | **0.02** | 0.01 | 0.01 | **0.44** | **0.06** | 0.01 | **0.06** | 0.02 |
| Ny-Ålesund | **0.26** | **0.16** | **0.15** | 0.08 | 0.09 | 0.00 | 0.00 | 0.02 | 0.00 | 0.12 |
| *$O_3$, ppb:* | | | | | | | | | | |
| Barentsburg | 35.74 | **39.59** | **45.26** | **36.74** | 35.33 | **45.76** | 35.41 | 25.05 | **41.84** | 36.41 |
| Zeppelin | **37.76** | 36.28 | **43.29** | **38.83** | 37.28 | **41.69** | 33.57 | 32.07 | **39.84** | 37.68 |
| wind speed, $m \cdot s^{-1}$: | | | | | | | | | | |
| Adventdalen | 3.9 | 7.0 | 4.9 | 7.8 | 4.1 | 4.6 | 3.3 | 4.6 | 3.2 | 4.6 |
| Barentsburg | 3.2 | 1.4 | 1.7 | 1.2 | 1.0 | 3.3 | 2.1 | 2.3 | 2.7 | 2.2 |
| Ny-Ålesund | 3.9 | 1.5 | 2.4 | 3.0 | 1.6 | 4.0 | 3.2 | 3.2 | 4.5 | 3.2 |
| wind direction: | | | | | | | | | | |
| Adventdalen | ESE, SE, E | SE, ESE, E | SE, ESE, E | SE, ESE, SSE | ESE, SE, SSE | W, SW, SSW | ESE, WNW, SW | ESE, SE, E | ESE, W, SE | ESE, SE, E |
| Barentsburg | E, ESE, NE | E, SE, SSE | SE, E, ESE | SSE, SE, S | SSE, SE, S | SSW, S, WSW | S, SSW, ENE | S, NE, ENE | S, SSE, SSW | SSE, S, E |
| Ny-Ålesund | SE, SSE, NNW | SSE, WSW, SE | SE, SSE, ESE | SE, SSE, ESE | SSE, WSW, S | WSW, W, WNW | SSE, WSW, WNW | SE, SSE, SW | W, SE, ESE | SE, SSE, WSW |
| temp. at 2 m, $^oC$: | | | | | | | | | | |
| Adventdalen | -15.1 | -6.6 | -8.9 | -10.6 | -11.4 | -1.1 | -2.4 | -9.1 | -1.2 | -9.9 |
| Barentsburg | -14.5 | -6.5 | -6.8 | -9.5 | -8.4 | -1.5 | -2.2 | -8.9 | -1.8 | -8.4 |
| Ny-Ålesund | -15.1 | -7.5 | -6.4 | -8.6 | -8.2 | -3.2 | -2.9 | -7.9 | -1.5 | -8.2 |
| PBL, m | | | | | | | | | | |
| Adventdalen | 299.4 | 252.3 | 251.1 | 270.5 | 146.2 | 635.4 | 366.3 | 373.4 | 269.6 | 298.2 |
| Barentsburg | 366.3 | 191.1 | 256.6 | 319.1 | 277.4 | 616.0 | 447.1 | 503.7 | 526.6 | 379.2 |
| Ny-Ålesund | 182.9 | 54.2 | 155.3 | 150.1 | 117.7 | 501.8 | 243.2 | 208.0 | 677.8 | 185.7 |
| UV-A+UV-B, $W \cdot m^{-2}$ | | | | | | | | | | |
| Longyearbyen | 1.4 | 3.4 | 3.1 | 4.7 | 4.3 | 4.9 | 3.5 | 8.6 | 5.2 | 3.8 |
| Ny-Ålesund | 2.4 | 6.1 | 5.1 | 7.9 | 5.4 | 7.5 | 7.7 | 13.1 | 7.1 | 6.3 |

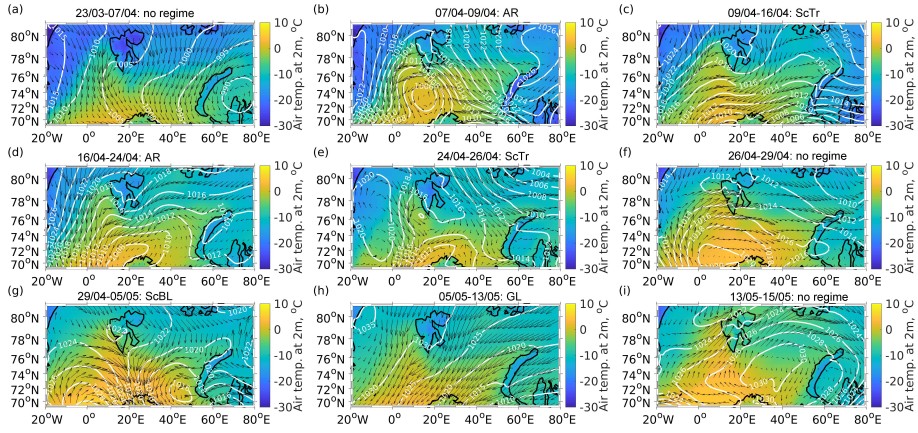

**Figure 8.** Synoptic scale meteorological conditions in ERA5 reanalysis data for the nine sub-periods. The color scale and the white contour lines show air temperature at two meter height and mean sea level pressure, respectively. The black arrows represent the prevailing wind direction and have the length relative to the wind speed. They are plotted with resolution of 2° of longitude and 1° of latitude.

The $O_3$ data show similar variability for the Zeppelin station and Barentsburg. The concentrations below the campaign's median were observed for the sub-periods V, VII and VIII. The FLEXPART 10 days backward trajectory probability contours show that for these sub-periods, the air masses passed over the region north of Svalbard where the concentration of BrO was elevated (Figure 9e), g), h)). On the contrary, the sub-periods III, IV, VI and IX, with $O_3$ concentration above median at both stations, are characterized by the air masses arriving from the south-east, east, west and south-west (Figure 9c), d), i)), respectively. In the sub-period I, 24% of the data from Barentsburg were missing. This sub-period's concentrations at the Zeppelin station were slightly higher than the campaign's median, despite the most significant $O_3$ depletion episode that occurred on the 31st of March- 1st of April (Figure 2). The trajectory contours show possible influence of the local depletion in the Svalbard region (Figure 9a). In the sub-period II, 67% of the data from the Zeppelin station were missing (Figure 2). The $O_3$ concentration in Barentsburg was above median for this sub-period, and the trajectory data show the air masses arriving from the south-east (Figure 9b). As in previous studies of Hirdman et al. (2009), the downward transport of $O_3$-enriched air masses from higher altitudes played significant role during the 2017 campaign. The percentage of trajectory points reaching elevations above 2000 m was highest for the sub-periods III, VI and IX (27%, 33% and 24% of the total number trajectory points for each sub-period respectively). In contrast, during the sub-period VIII with the lowest $O_3$ concentration at both stations, the percentage of elevated trajectory points was minimal, only 4%. One can also see that the percentage of elevated trajectories varies for the same type of weather regime and determines importance of the downward air mass transport for the measured surface $O_3$ concentrations in different sub-periods (e.g. ScTr regime in Figure 9c) and e) and Table2).

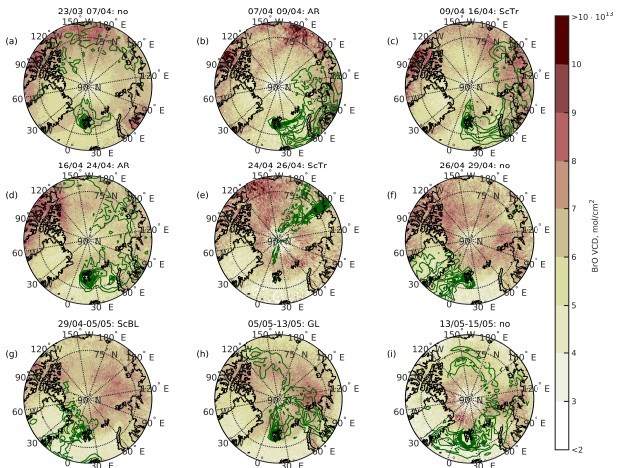

**Figure 9.** FLEXPART trajectory probability for 10 days backward trajectories (dark green contours with step of 0.001) and GOME2 BrO vertical column density (VCD) (color scale) for the different sub-periods. The percentage of trajectories descending from higher altitudes (>2000 m) is 7%, 12%, 27%, 9%, 9%, 33%, 18%, 4% and 24% for the sub-periods in plots (**a**)-(**i**), respectively.

Two joint extreme $O_3$ depletion events (31.03.2017 and 06.05.2017) and three increase events (13.04.2017, 28.04.2017 and 03.05.2017) have been detected (Figures 2 and 3). The HYSPLIT trajectory analysis shows that these $O_3$ depletion events occurred when the cold air masses from the central Arctic reached Svalbard. The trajectory for the strongest depletion episode is shown in Figure 10a). The concentration of $O_3$ in the Arctic air masses may be lower because of lack of sunlight and $O_3$ precursors such as $NO_x$, hydrocarbons and CO needed for the $O_3$ formation. Further depletion may have occurred due to photochemical reactions with bromine species over the sea ice in the period from 30.03.2017 10:00 to 31.03.2017 17:00 when the trajectories passed the region with elevated BrO concentration between $80^oN$ and $85^oN$. The simulated median sun flux was quite low ($67\ W \cdot m^{-2}$), but probably sufficient enough to support the halogen-induced $O_3$ destruction, which might occur even under low-light conditions (Simpson et al., 2015). The trajectories for the increase events revealed southerly origin of the air masses, but source regions were different for all three cases. In the first case, air masses were arriving from Northern part of Russia, and in the second one from North America and Iceland. However, the highest $O_3$ concentrations at both stations were observed 03.05.2017 when the air masses were transported from Europe (Figure 10b)). The air masses arrived to Svalbard from the west and did not pass over the areas with elevated BrO concentration.

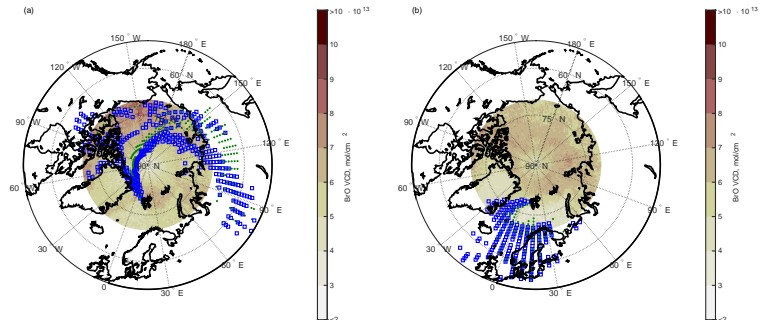

**Figure 10.** HYSPLIT 10-days air mass backward trajectories probability for the strongest O$_3$ depletion (**a**) and O$_3$ increase (**b**) events detected both in Barentsburg (green dots) and at the Zeppelin station (blue squares). The points show the trajectory probability above median calculated for the ensemble with 27 trajectories. The 10-days mean BrO total VCD for the Arctic region ($>70^o N$) is shown with the color scale. The percentage of trajectories descending from higher altitudes ($>2000$ m) is 1% and 3% for the depletion case (**a**) and 25% and 14% for the O$_3$ increase (**b**) for Zeppelin and Barentsburg, respectively.

## 4    Discussion

The NO$_x$ monitor in Adventdalen was located far away from stationary emission sources and showed the highest daytime NO$_2$/NO$_x$ ratio (Table 1). We would like to investigate how the ratios observed there were affected by photolysis. The photol-
ysis rate of NO$_2$ depends on solar zenith angle (Parrish et al., 1983), which in turn depends on day of year. Measurements were performed between days 81 and 134, and the noon solar zenith angle in Longyearbyen area varied from approximately 77° to 62° (Robertson et al., 2006). Following equation (15) in Parrish et al. (1983), minimum clear-sky photolysis rate for the start of the campaign was 0.0026 s$^{-1}$, and maximum clear-sky photolysis rate for the end of the campaign was 0.0061 s$^{-1}$ (black squares in Figure A1a)). There are many factors that affect NO$_2$ photolysis rate, such as aerosol load, clouds, water vapour
content and surface albedo (Trebs et al., 2009). The albedo may significantly increase the NO$_2$ photolysis rate (Trebs et al., 2009), and Dickerson et al. (1982) suggested albedo of snow with respect to j(NO$_2$) to be 93%. Trebs et al. (2009) suggested in their equation (2) a polynomial fit between global irradiance and NO$_2$ photolysis rate that includes both clear-sky and cloudy conditions and takes into account the contribution of albedo. The albedo calculated as the ratio of upward and downward short-wave radiation measured by CNR1 Kipp Zonen net radiometer in Adventdalen and observed global radiation were used
to estimate j(NO$_2$) (red line in Figure A1a)). Figure A1b) shows NO/NO$_2$ ratio calculated using O$_3$ concentration measured in Barentsburg (closest station where O$_3$ measurements were available) , j(NO$_2$) and temperature-dependent rate coefficient k$_{NO+O_3}$ obtained using temperatures in Adventdalen (equation 6.6 and Table 6.1 in Seinfeld and Pandis (2006)). The peaks of NO/NO$_2$ ratios are especially pronounced for the days with decreased O$_3$ concentration (01.04.2017 and the period from 04.05.2017 to 09.05.2017). Note, the calculation is based on the O$_3$ data from Barentsburg, thus this introduces an uncer-
tainty in the exact NO/NO$_2$ ratios estimated for Adventdalen. The observed and calculated NO$_2$/NO$_x$ ratio for Adventdalen

are shown in Figure A1c). The missing data in the observed $NO_2/NO_x$ ratio (light blue line) indicate that both NO and $NO_2$ values were within zero-noise level, while missing data in the calculated $NO_2/NO_x$ ratio is due to missing $O_3$ observations in Barentsburg. The observed and calculated values are of the same order, but $NO_2/NO_x$ ratio is underestimated in 64% of all available data, especially for the days with low $O_3$ values. This underestimation was present even in hours influenced by fresh local NO emission (light purple line) and might have resulted from the modelling errors that could occur if the surface albedo was high (Trebs et al., 2009) or because the actual $O_3$ values in Adventdalen were lower than in Barentsburg. The $NO_2/NO_x$ ratio is overestimated in 31% of all available data. In these hours, the actual $O_3$ concentration might have been higher in Adventdalen than in Barentsburg (used for calculations). The most pronounced overestimation is noticeable in the period from 26.04 to 29.04 when NO values in Barentsburg were higher than in Adventdalen, and thus more pronounced $O_3$ titration with local NO might have occurred in Barentsburg.

The results from radiosonde and ozone soundings, CO and particle measurements, presented in this study, demonstrate that the $O_3$ observations at the Zeppelin station were not sensitive to the local $NO_x$ pollution from Ny-Ålesund, and thus were representative as background values for comparison with Barentsburg and investigation of the influence of prevailing long-range transport patterns on the measurements at these stations. Furthermore, $O_3$ data from the Zeppelin station may be used to assess how the PAN decomposition might have affected the background $NO_x$ concentrations in Svalbard during the 2017 campaign. Previous studies have shown that the $NO_x$/PAN ratio increases at temperatures above -10 °C, and PAN decomposition becomes a major source of background $NO_x$ in Svalbard (Beine et al., 1997a; Beine and Krognes, 2000). As the temperature at the Zeppelin station varied from -22.7 to 0.8 °C during the campaign, we would like to investigate the contribution of PAN decomposition to the background $NO_x$ concentration in Svalbard. The PAN decomposition rate may be estimated using several approaches (Beine et al., 1997a), here we apply a linear relationship between $O_3$ and PAN concentration derived from previous measurements at the Zeppelin station: PAN[ppt]=($O_3$[ppb]-26.58)/0.034 and then calculate PAN decomposition rate (Beine and Krognes, 2000) (Figure A2a). The maximum PAN decomposition rate has been calculated using temperatures and $O_3$ concentration observed at the Zeppelin station applying equation (1) from Beine and Krognes, 2000 (Beine and Krognes, 2000) . The depletion events when $O_3$ concentration was below 26.58 ppb have been excluded from the calculation (Beine and Krognes, 2000). The median calculated PAN concentration of 356 pptv (ppb x$10^{-3}$) is comparable with previous springtime Arctic observations (Beine and Krognes, 2000; Kramer et al., 2015) .The estimated maximum PAN decomposition rate for the whole campaign varied from -0.0033 to -17.2 pptv $hour^{-1}$ with median value of -1.29 pptv $hour^{-1}$ (Figure A2b). The maximum PAN concentration coincides with the strongest $O_3$ increase event occurred 03.05.2017 (Figure 10b). The temperature increased simultaneously for that day (Figure 2a) promoting efficient PAN decomposition (Figure A2b). Applying Theil's non-parametric regression with the slope of –5.07 (pptv $NO_x$/pptv $hour^{-1}$ PAN) suggested by Beine et al. (1997a) for Svalbard, the background concentration of $NO_x$ would be 87.2 pptv. However, these concentrations are too low for the equipment used in the 2017 campaign to detect the variations in the concentrations caused by the PAN decomposition.

The absence of collocated $NO_x/O_3$ measurements in Ny-Ålesund and Longyearbyen does not allow to investigate how the local emissions affect $O_3$ concentrations in these settlements. This is a drawback of this study. The $O_3$ monitoring at the Zeppelin station is a long-term ongoing research project, and relocation of the instrument to the village from the mountain

observatory would introduce bias in the long-term atmospheric composition observations. The study in Adventdalen was the first combined air pollution and meteorological field work in Longyearbyen. The measurements there were done by the main author, and only $NO_x$ monitor was installed there due to the limited grant funding.

To investigate the influence of local $NO_x$ emissions on the $O_3$ concentration in Ny-Ålesund, as it is required in the third hypothesis stated in the introduction of the current paper, we may use historical observations. The data from only six $O_3$ sonde launches were available for the 2017 campaign (Figure 4). However, the long-term data below 100 m from the $O_3$ sonde profiles may be used to study influence of the local $NO_x$ pollution in Ny-Ålesund on the $O_3$ concentration. These observations are suitable for this purpose because the $O_3$ sonde launching facility is located just 200 m to the south-south-west and 500 m to the south from the $NO_x$ monitor and diesel power plant, respectively. Thus, when the monitor detected $NO_x$ concentration above long-term springtime average in the launch hour, the influence of locally polluted air masses might have been observed in the lowest $O_3$ sonde data. There were in total 59 $O_3$ sonde launches, for which $NO_x$ monitor data was available in spring 2009, 2010, 2015, 2016, 2017 and 2018. The $O_3$ profile data in the lowest 100 m have been extracted for all 59 launches and grouped according to the $NO_x$ concentration detected by the monitor and wind direction in the $O_3$ sonde profiles: 1) above mean $NO_x$ concentration and northerly wind direction; 2) below or equal to mean $NO_x$ concentration and northerly wind direction. The median and mean $O_3$ values below 100 m in the group where the $NO_x$ values were above $NO_x$ mean were 11 % and 15% lower, respectively, than for the second group with northerly winds, but without elevated $NO_x$ concentration. Thus, the $O_3$ concentration in lowest 100 m downwind from the power plant in the settlement may be reduced significantly due to local $NO_x$ emissions, but the frequency of such events is unknown in absence of continuous $O_3$ measurements in the village.

The $NO_x$ concentrations depended strongly on the wind direction at the stations located in the vicinity if the stationary pollution sources: Ny-Ålesund and Barentsburg. In turn, the wind direction at all the sites depended on the synoptic-scale conditions, but was modified locally due to different mechanical and thermodynamic processes controlling local circulations such as katabatic winds and topography-induced wind channelling specific for each location (Esau and Repina, 2012; Maturilli et al., 2013). Remarkably, westerly component of the wind at all stations only appears when the synoptic scale westerly wind brought warm air from North Atlantic to the Svalbard inland during transition to, and during part of the large-scale ScBL regime. This reverses semi-permanent thermal flow from the glaciers towards the sea prevailing in Svalbard in spring.

Our analysis of the trajectory probability for different weather regimes showed that the elevated median $O_3$ concentrations were observed for the sub-periods when the air masses arrived from the south, west or east. In contrast, the long-range transported $O_3$, brought by the air masses from the north, may be affected by the regional $O_3$ depletion north of Svalbard where the elevated concentrations of total BrO VCDs were detected in the satellite data. Similarly, Koo et al. (2012) studied advection of the $O_3$-depleted air masses and found that these events were driven by local or short-range (1 day) transport from the nearby region. Recent study of Bougoudis et al. (2020) explored connection between first-year sea ice and bromine explosion events. In spring (March, April, May) 2017, Arctic mean tropospheric BrO VCDs over sea ice were on the order of $4 \cdot 10^{13} molecules \cdot cm^{-2}$, and significant anomalies of tropospheric BrO VCDs were observed over the sea ice north of Svalbard at approximately $85^o$N. In addition to the sea ice conditions, the tropospheric BrO plumes formation depend on various meteorological factors and the amounts of blowing snow (Bougoudis et al., 2020). To investigate these processes, the weather

regime approach presented in the current study may be applied together with the long-term BrO remote sensing data and *in-situ* measurements from the Arctic stations in further studies.

However, as we show in the analysis of the extreme $O_3$ depletion and increase episodes, the specific short-term events of long-range transport need to be investigated separately as there is a spread in air mass origin and transport altitude for the longer periods defined by the weather regime classification, especially for the "no regime" situation.

To get a more robust result linking weather regimes and air quality, we would like to compare long-term springtime (23 March-15 May) weather regime data with $NO_x$ data from Ny-Ålesund, $O_3$ concentration from the Zeppelin station, $O_3$ sonde and radiosonde data as well as FLEXPART trajectories for a period from 1990 to 2018. The FLEXPART and weather regime data were available for all years, while there were gaps in observational data from Ny-Ålesund. The data availability chart is shown in Figure A3 indicating number of hourly measurements for surface $NO_x$ and $O_3$ data and number of radiosonde and ozone sonde launches per spring season each year. The hourly $O_3$ data is available for all years, while hourly $NO_x$ data was only available in 2009, 2010, 2015, 2016, 2017 and 2018. After spring 2018, the $NO_x$ monitor was moved to other location in Ny-Ålesund, therefore 2019 - 2022 data are not included in the current analysis to keep measurement consistency. The $O_3$ soundings and radiosonde AWI's datasets start in 1992 and 1993 with median number of radiosonde soundings and $O_3$ soundings per spring season being 54 and 11, respectively.

The box and whisker plots of $O_3$, $NO_x$ and temperature inversion strength (TIS) for different weather regimes are shown in Figure A4. As during the 2017 campaign, the highest median $O_3$ values were detected for ScTr regime, while Zonal regime (ZO), during which $O_3$ values were also higher, was absent during the field work (Figure A4a)). The ZO regime is characterized by negative geopotential height anomaly at 500 hPa centred between Iceland and southern tip of Greenland and southerly flow over Svalbard (Papritz and Grams, 2018). The lowest median $O_3$ values in the long-term data are for GL regime and for European blocking (EUBL) that was absent in spring 2017. The variability of $O_3$ concentrations (range between the $25^{th}$ percentile and $75^{th}$ percentile) for the EUBL regime is also remarkably higher than for other regimes. The EUBL regime is characterized by the positive 500 hPa anomaly centred over the North Atlantic. This promotes transport of air from south-west and west to Svalbard (Papritz and Grams, 2018). Long-term trajectories for the different regimes are shown in Figure A5. The lowest percentage of trajectories descending from higher altitude (>2000 m) is modelled for GL regime, while the highest percentage of elevated trajectories is obtained for EUBL, ScBL and ZO regimes. No specific trajectory probability pattern may be defined for "no regime" (Figure A5a), while distinct long-range transport signatures are identified for other seven regimes (Figure A5b)-h)). In addition to the air transport path and trajectory altitudes, the sea-ice conditions and BrO concentration are important factors affecting the concentration of $O_3$ in each particular season.

Similar to the 2017 results, the highest median $NO_x$ values are observed for "no" regime, ScTr, but two other regimes, when the long-term median $NO_x$ values are high as well, EUBL and ZO, were not present during the campaign (Figure A4b)). The EUBL show pronounced transport of air masses from the west to Svalbard (Figure A5f). Thus, a westerly component of the wind at the measurement stations and significant changes in local pollution dispersion conditions are expected for this regime as was observed during ScBL regime in 2017 when westerly component of the wind was present as well.

Temperature inversions are common phenomena at high latitudes, in particular during the cold seasons due to radiative cooling of the surface and descending motion and heat advection from the south aloft. The inversions were detected in 27% of all the days in the measurement campaign period in 2017. This frequency of inversion occurrence is quite low in comparison with the results from previous studies of Dekhtyareva et al. (2018), where it was observed in 60% of the springtime profiles in 2009. Despite low frequency of occurrence, temperature inversions have significant influence on the dispersion efficiency, and, hence, according to the WRS-test, the median daytime (from 06UTC to 18UTC) concentrations of $NO_x$ were higher ($p<0.05$) at all three stations for the days when this phenomenon was observed in the radiosonde data in 2017. In the long-term data, the median temperature inversion strength was high for no regime, GL, AR and ScBL, but the highest median TIS was for Atlantic trough (AT) regime, the regime that was absent during the campaign (Figure A4c)). The AT regime is characterized by a negative 500 hPa geopotential height anomaly to the east of Ireland and high cyclone frequency in that region (Papritz and Grams, 2018), while the cyclonic activity around Svalbard is lower, and these conditions may promote strengthening of the temperature inversion.

Thus, the results of the weather regime analysis performed for 2017 campaign are representative for characterization of the influence of different synoptic scale conditions on the $NO_x$ and $O_3$ concentration in Ny-Ålesund. However, the three regimes that were absent during the 2017 campaign (AT, EUBL and ZO) are important in the long-term statistics for $NO_x$, $O_3$ and TIS in the settlement.

## 5 Conclusions

Despite decades of industrial activity in Svalbard, there is no continuous air pollution monitoring in the region's settlements except Ny-Ålesund. The $NO_x$ measurement results from the three stations-network, Ny-Ålesund, Barentsburg and Longyearbyen, and $O_3$ data from Ny-Ålesund and Barentsburg have been compared for the first time.

A diurnal pattern in concentration of $NO_x$ at all three stations has been observed and attributed to variable emissions from the local sources of $NO_x$. However, only data from Barentsburg and Adventdalen station show significant change in $NO_2/NO_x$ ratio during the day, since the station in Ny-Ålesund is located close to a diesel power plant, a stationary source of fresh $NO_x$ emissions contributing to higher NO concentration. The $NO_2/NO_x$ ratio observed in Adventdalen is comparable with modelling results obtained using radiation data from the valley and $O_3$ concentration from Barentsburg. Local emissions of $NO_x$ in Barentsburg may reduce $O_3$ concentrations in the settlement by a few percent from the background value due to $O_3$ titration. As it has been shown from the analysis of long-term $O_3$ sonde record, $NO_x$ emissions in Ny-Ålesund may affect $O_3$ concentration in the lowest 100 m downwind, but no influence of local pollution has been detected at the Zeppelin station at 474 m a.s.l. in spring 2017. There was no statistically significant difference in daytime and nighttime $O_3$ values measured in Barentsburg and at the Zeppelin station, and both sites showed similar $O_3$ concentration dynamics controlled by long-range air mass transport.

The weather regime approach is novel in Svalbard air pollution research. This method has been used in the current study to identify the influence of large-scale circulation on local and long-range transported air pollution in the area with complex

topography. As expected, the large-scale wind is channelled by the local topographical features and this determines the wind direction and speed in all three settlements, and therefore the correlations of $NO_x$ concentrations between the stations are weak.

In Ny-Ålesund and Barentsburg, the stations are located so that downwind concentrations from the local sources are observed rarely, since the prevailing wind direction is different. The measurements in Adventdalen have been made downwind from the source, since both the snowmobile route and prevailing wind direction are along the valley. However, traffic is a temporary source of emissions and the mean wind speed in Adventdalen valley is high, and therefore mean $NO_x$ concentrations there are low. Despite low correlation between the $NO_x$ values from the three stations, there are common synoptic conditions that promote accumulation of local pollution in the settlements, namely, low wind speed and air temperature and presence of temperature inversions. In contrast to $NO_x$, the concentrations of $O_3$ in Barentsburg and at the Zeppelin observatory are moderately correlated and depend on synoptic conditions that promote transport of air masses enriched or depleted in $O_3$. In other words, both these stations are regionally representative for the $O_3$ concentrations.

The large-scale weather regimes control the synoptic meteorological conditions and determine the atmospheric stability and efficiency of local pollution dispersion. The analysis of long-term weather regime, trajectory and observational data from Ny-Ålesund, support our findings from 2017 campaign. The lowest median $O_3$ values were identified for Greenland blocking regime, when the trajectories were arriving more frequently from the sea-ice covered regions and the percentage of high-altitude trajectories that might bring $O_3$-enriched air was low. The highest median $O_3$ values were observed for Scandinavian trough regime, characterized by the relatively high percentage of high-altitude trajectories and trajectories arriving from south-east of Svalbard. During this regime, $NO_x$ concentrations were elevated as well. The highest temperature inversion strength was observed for "no regime", Greenland blocking, Atlantic ridge and Scandinavian blocking. The maximum background $NO_2$ concentration originating from PAN decomposition was modelled for the strongest $O_3$ increase event occurred during Scandinavian blocking regime. At the same time, three regimes absent during the 2017 campaign (Atlantic trough, European blocking and Zonal regime) appeared to be significant for $NO_x$, $O_3$ and temperature inversion statistics in Ny-Ålesund. However, the effect of each weather regime on the air quality in different settlements depends on the local features such as pollution sources and wind channeling, thus it would be of interest to compare long-term $NO_x$ and $O_3$ data from Barentsburg with weather regime data in further studies.

The application of the weather regime approach in air quality study for the three Svalbard stations allows to facilitate prediction of the conditions promoting long-range transport to and accumulation of local pollutants at the measurement sites. The weather regimes typically persist for a period of 10 days and longer. Hence, joint intensive observational campaigns may be planned ahead at any of the three stations depending on the expected conditions. This provides a new opportunity for the collaboration in atmospheric research in Svalbard and allows more effectively organise specific field observations devoted to, for example, study of photochemical reactions in polar atmosphere, investigations of influence of turbulence and stability on air pollutant dispersion, studies of aerosol and cloud interaction.

*Data availability.* The radiosonde data for March 2017 and for April-May 2017 are available via the GRUAN homepage www.gruan.org and
in the database www.pangaea.de (Maturilli, 2017a), (Maturilli, 2017b), respectively. The analysed $O_3$ sonde data are stored in the Network for
the Detection of Atmospheric Composition Change (NDACC) archive ftp://ftp.cpc.ncep.noaa.gov/ndacc/station/nyalsund/ames/$O_3$sonde/.
The $NO_x$ data from Adventdalen are available at the UiT open research data portal (Dekhtyareva, 2018).

The following authors' contributions have been made: conceptualization, D.A.; methodology, D.A., G.R., H.K. and H.M.;
software, D.A. and G.R.; validation, D.A. and G.R.; formal analysis, D.A., G.R. and S.T.; investigation, D.A., H.O., H.M., S.T.
and N.A.; resources, D.A., H.K., N.A., H.O. and H.M.; data curation, D.A., S.T., N.A., H.O., H.M.; writing—original draft
preparation, D.A.; writing—review and editing, D.A., H.K., G.R., S.T., H.M.; visualization, D.A.; supervision, D.A.; project
administration, D.A.; funding acquisition, D.A., H.K., N.A. and H.O.

The measurements of $NO_x$ in Adventdalen have been performed in frame the project 269953/E10 «Monitoring of nitrogen
oxides from mobile and stationary sources at Svalbard» financed by the Arctic Field Grant funding established by Norwegian
Research Council. The measurements of $NO_x$ are performed by the NPI and NILU with a logistical support from Kings
Bay AS in connection with the project «Limits of Acceptable Change» in Ny-Ålesund. Continuous $O_3$ measurements at the
Zeppelin station are performed in the frame of the long-term programme for greenhouse gases monitoring and financed by
NILU and Norwegian Environmental Agency. The measurements in Barentsburg have been done by AARI in the scope of
the project «Air quality monitoring by automatic analysing stations in Barentsburg». The support from the Transregional
Collaborative Research Centre (TR 172) "ArctiC Amplification: Climate Relevant Atmospheric and SurfaCe Processes, and
Feedback Mechanisms (AC)3," funded by the German Research Foundation (DFG, Deutsche Forschungsgemeinschaft) is
acknowledged for the radiosonde data from Ny-Ålesund.

The authors declare no conflict of interest. The funders had no role in the design of the study; in the collection, analyses, or
interpretation of data; in the writing of the manuscript, or in the decision to publish the results.

*Acknowledgements.* Special thanks are given to the staff of the Norwegian Polar Institute (NPI) and the University centre in Svalbard for
the invaluable logistical assistance. Norwegian Institute for Air Research is acknowledged for the leasing of the equipment and technical
support during the operation of the monitor. We would like to acknowledge Norwegian meteorological institute for the meteorological data
from Ny-Ålesund available in the eklima.no database. NPI and the European Centre for Medium-Range Weather Forecasts are acknowledged
for the map of Svalbard available at http://svalbardkartet.npolar.no and for the data from the ERA5 global atmospheric reanalysis data set,
accordingly. We would like to thank Dr. Marion Maturilli and Dr. Peter von der Gathen from the Alfred Wegener Institute Helmholtz Centre
for Polar and Marine Research for processing and quality assurance of the radiosonde and $O_3$ sonde data from Ny-Ålesund, respectively, and
for the reviewing of the early version of the current paper. Special appreciation is given to Dr. Christian Grams from the Karlsruhe Institute
of Technology for providing the weather regime data and useful comments for the current paper.

no

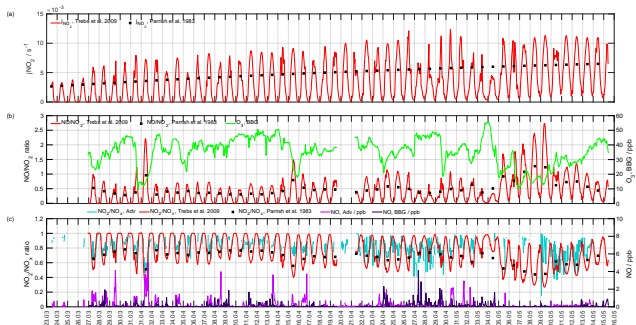

**Figure A1.** (**a**) $j_{NO_2}$ calculated as a function of solar zenith angle in clear-sky conditions (Parrish et al., 1983) (decreasing from 77° to 59° for solar noon time in Adventdalen from 23.03.2017 to 15.05.2017 (Robertson et al., 2006)) and as a function of observed global radiation and albedo (Trebs et al., 2009); (**b**) $NO/NO_2$ ratio calculated using $O_3$ concentration measured in Barentsburg, $j_{NO_2}$ and temperature dependent rate coefficient $k_{NO+O_3}$ obtained using temperatures in Adventdalen (equation 6.6 and Table 6.1 in (Seinfeld and Pandis, 2006)); (**c**) observed and calculated $NO_2/NO_x$ ratio for Adventdalen.

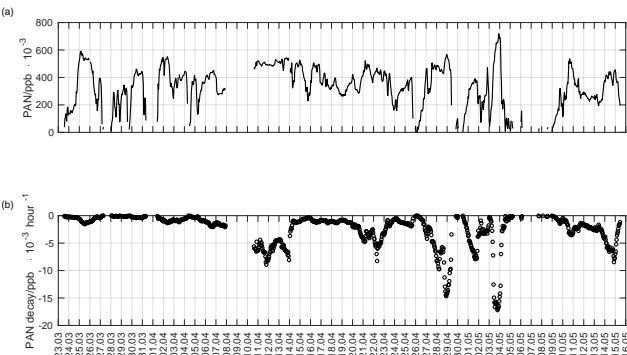

**Figure A2.** (**a**) PAN concentration calculated using linear relationship between PAN and $O_3$ concentration suggested by (Beine and Krognes, 2000); (**b**) PAN decomposition rate calculated using equations 1 and 2 from (Beine and Krognes, 2000).

**Appendix A**

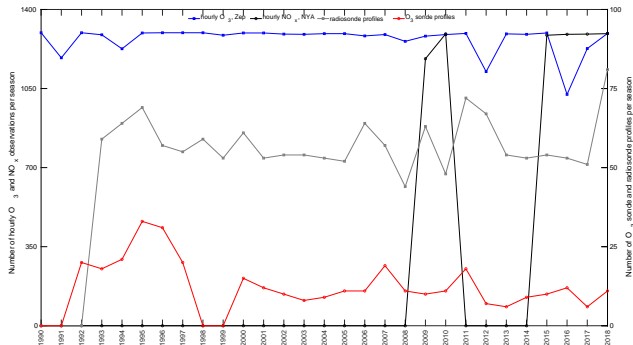

**Figure A3.** The $O_3$ and $NO_x$ hourly observations and $O_3$ sonde and radiosonde data availability chart for spring seasons 1990-2018.

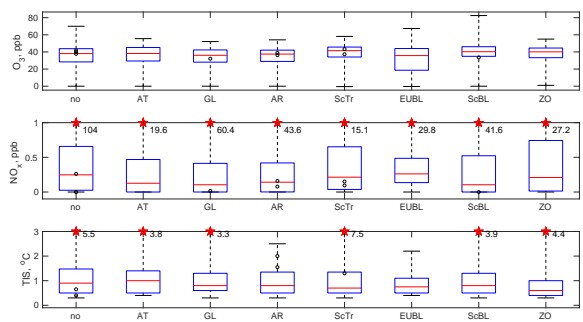

**Figure A4.** (**a**) $O_3$ concentrations for different weather regimes for spring seasons 1990-2018; (**b**) $NO_x$ concentrations for different weather regimes for spring seasons 1990-2018; (**c**) Temperature inversion strength for different weather regimes for spring seasons 1990-2018. The maximum values that exceed y-axis limits in plots (**b**) and (**c**) are shown with red stars.

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

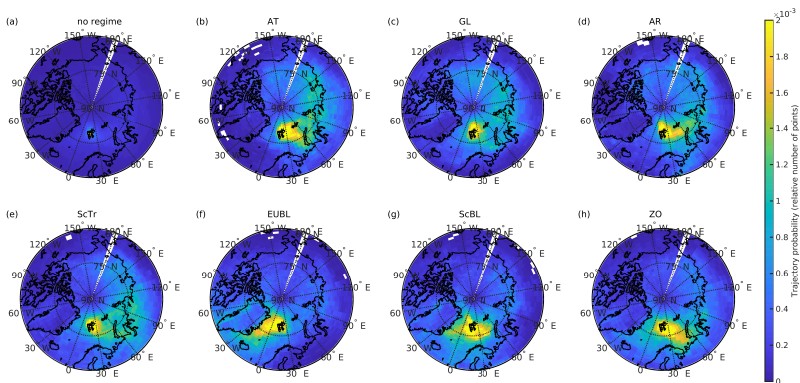

**Figure A5.** FLEXPART trajectory probability for 10 days backward trajectories for the different weather regimes for spring seasons 1990-2018. The percentage of trajectories descending from higher altitudes (>2000 m) is 18%, 14%, 13%, 15%, 17%, 22%, 21% and 21% for the regimes in plots (**a**)-(**h**), respectively.

Beine, H. J., Engardt, M., Jaffe, D., Hov, Ø., Holmén, K., and Stordal, F.: Measurements of NOx and aerosol particles at the Ny-Ålesund Zeppelin mountain station on Svalbard: influence of regional and local pollution sources, Atmospheric Environment, 30, 1067–1079, 1996.

Beine, H. J., Jaffe, D. A., Herring, J. A., Kelley, J. A., Krognes, T., and Stordal, F.: High-Latitude Springtime Photochemistry . Part I : NOx, PAN and Ozone Relationships, Journal of Atmospheric Chemistry, 27, 127–153, 1997a.

Beine, H. J., Jaffe, D. A., Stordal, F., Engardt, M., Solberg, S., Schmidbauer, N., and Holmén, K.: NOx during ozone depletion events in the arctic troposphere at Ny-Ålesund, Svalbard, Tellus, Series B: Chemical and Physical Meteorology, 49, 556–565, https://doi.org/10.3402/tellusb.v49i5.16008, 1997b.

Bougoudis, I., Blechschmidt, A. M., Richter, A., Seo, S., Burrows, J. P., Theys, N., and Rinke, A.: Long-term time series of Arctic tropospheric BrO derived from UV-VIS satellite remote sensing and its relation to first-year sea ice, Atmospheric Chemistry and Physics, 20,

11 869–11 892, https://doi.org/10.5194/acp-20-11869-2020, 2020.

Christiansen, B., Jepsen, N., Kivi, R., Hansen, G., Larsen, N., and Smith Korsholm, U.: Trends and annual cycles in soundings of Arctic tropospheric ozone, Atmospheric Chemistry and Physics, 17, 9347–9364, https://doi.org/10.5194/acp-17-9347-2017, 2017.

Dee, D. P., Uppala, S. M., Simmons, A. J., Berrisford, P., Poli, P., Kobayashi, S., Andrae, U., Balmaseda, M. A., Balsamo, G., Bauer, P., Bechtold, P., Beljaars, A. C. M., van de Berg, L., Bidlot, J., Bormann, N., Delsol, C., Dragani, R., Fuentes, M., Geer, A. J., Haim-

720 berger, L., Healy, S. B., Hersbach, H., Hólm, E. V., Isaksen, L., Kållberg, P., Köhler, M., Matricardi, M., McNally, A. P., Monge-Sanz, B. M., Morcrette, J.-J., Park, B.-K., Peubey, C., de Rosnay, P., Tavolato, C., Thépaut, J.-N., and Vitart, F.: The ERA-Interim reanalysis: configuration and performance of the data assimilation system, Quarterly Journal of the Royal Meteorological Society, 137, 553–597, https://doi.org/10.1002/qj.828, 2011.

Dekhtyareva, A.: Monitoring of nitrogen oxides at Svalbard: measurements in Adventdalen, https://doi.org/10.18710/TXQ7EV, 2018.

Dekhtyareva, A., Edvardsen, K., Holmén, K., Hermansen, O., and Hansson, H. C.: Influence of local and regional air pollution on atmospheric measurements in Ny-Ålesund, International Journal of Sustainable Development and Planning, 11, 578–587, https://doi.org/10.2495/SDP-V11-N4-578-587, 2016.

Dekhtyareva, A., Holmén, K., Maturilli, M., Hermansen, O., and Graversen, R.: Effect of seasonal mesoscale and microscale meteorological conditions in Ny-Ålesund on results of monitoring of long-range transported pollution, Polar Research, 37, 1508 196, https://doi.org/10.1080/17518369.2018.1508196, 2018.

Dickerson, R. R., Stedman, D. H., and Delany, A. C.: Direct measurements of ozone and nitrogen dioxide photolysis rates in the troposphere., Journal of Geophysical Research, 87, 4933–4946, https://doi.org/10.1029/JC087iC07p04933, 1982.

Eckhardt, S., Stohl, A., Beirle, S., Spichtinger, N., James, P., Forster, C., Junker, C., Wagner, T., Platt, U., and Jennings, S. G.: The North Atlantic Oscillation controls air pollution transport to the Arctic, Atmospheric Chemistry and Physics, 3, 1769–1778, https://doi.org/10.5194/acp-3-1769-2003, 2003.

Eckhardt, S., Hermansen, O., Grythe, H., Fiebig, M., Stebel, K., Cassiani, M., Baecklund, A., and Stohl, A.: The influence of cruise ship emissions on air pollution in Svalbard ndash; A harbinger of a more polluted Arctic?, Atmospheric Chemistry and Physics, 13, 8401–8409, https://doi.org/10.5194/acp-13-8401-2013, 2013.

Esau, I. and Repina, I.: Wind climate in Kongsfjorden, Svalbard, and attribution of leading wind driving mechanisms through turbulence-resolving simulations, Advances in Meteorology, 2012, Article ID 568 454, https://doi.org/10.1155/2012/568454, 2012.

European Centre for Medium-Range Weather Forecasts: IFS DOCUMENTATION – Cy43r3 Operational implementation 11 July 2017 PART IV: PHYSICAL PROCESSES, Tech. Rep. July, https://www.ecmwf.int/sites/default/files/elibrary/2017/17736-part-iv-physical-processes.pdf, 2017.

Fan, S.-M. and Jacob, D. J.: Surface ozone depletion in Arctic spring sustained by bromine reactions on aerosols, Nature, 359, 522–524, 1992.

Førland, E. J., Hanssen-Bauer, I., and Nordli, P. Ø.: Climate statistics  longterm series of temperature and precipitation at Svalbard and Jan Mayen, Tech. rep., Norwegian Meteorological Institute, Oslo, 1997.

Fremme, A. and Sodemann, H.: The role of land and ocean evaporation on the variability of precipitation in the Yangtze River valley, Hydrology and Earth System Sciences, 23, 2525–2540, https://doi.org/10.5194/hess-23-2525-2019, 2019.

Freud, E., Krejci, R., Tunved, P., Leaitch, R., Nguyen, Q. T., Massling, A., Skov, H., and Barrie, L.: Pan-Arctic aerosol number size distributions: Seasonality and transport patterns, Atmospheric Chemistry and Physics, 17, 8101–8128, https://doi.org/10.5194/acp-17-8101-2017, 2017.

Grams, C. M., Beerli, R., Pfenninger, S., Staffell, I., and Wernli, H.: Balancing Europe's wind-power output through spatial deployment informed by weather regimes, Nature Climate Change, 7, 557–562, https://doi.org/10.1038/NCLIMATE3338, 2017.

Gröbner, J., Hülsen, G., Wuttke, S., Schrems, O., De Simone, S., Gallo, V., Rafanelli, C., Petkov, B., Vitale, V., Edvardsen, K., and Stebel, K.: Quality assurance of solar UV irradiance in the Arctic, Photochemical  photobiological sciences, 9, 384–391, https://doi.org/10.1039/b9pp00170k, 2010.

Heintzenberg, J., Tunved, P., Galí, M., and Leck, C.: New particle formation in the Svalbard region 2006 – 2015, Atmospheric Chemistry and Physics, 17, 6153–6175, https://doi.org/10.5194/acp-17-6153-2017, 2017.

Hersbach, H., Bell, B., Berrisford, P., Hirahara, S., Horányi, A., Muñoz-Sabater, J., Nicolas, J., Peubey, C., Radu, R., Schepers, D., Simmons, A., Soci, C., Abdalla, S., Abellan, X., Balsamo, G., Bechtold, P., Biavati, G., Bidlot, J., Bonavita, M., De Chiara, G., Dahlgren, P., Dee, D., Diamantakis, M., Dragani, R., Flemming, J., Forbes, R., Fuentes, M., Geer, A., Haimberger, L., Healy, S., Hogan, R. J.,

Hólm, E., Janisková, M., Keeley, S., Laloyaux, P., Lopez, P., Lupu, C., Radnoti, G., de Rosnay, P., Rozum, I., Vamborg, F., Villaume, S., and Thépaut, J. N.: The ERA5 global reanalysis, Quarterly Journal of the Royal Meteorological Society, 146, 1999–2049, https://doi.org/10.1002/qj.3803, 2020.

Hirdman, D., Aspmo, K., Burkhart, J. F., Eckhardt, S., Sodemann, H., and Stohl, A.: Transport of mercury in the Arctic atmosphere: Evidence for a springtime net sink and summer-time source, Geophysical Research Letters, 36, 1–5, https://doi.org/10.1029/2009GL038345, 2009.

Hirdman, D., Burkhart, J. F., Sodemann, H., Eckhardt, S., Jefferson, A., Quinn, P. K., Sharma, S., Ström, J., and Stohl, A.: Long-term trends of black carbon and sulphate aerosol in the Arctic: Changes in atmospheric transport and source region emissions, Atmospheric Chemistry and Physics, 10, 9351–9368, https://doi.org/10.5194/acp-10-9351-2010, 2010a.

Hirdman, D., Sodemann, H., Eckhardt, S., Burkhart, J. F., Jefferson, A., Mefford, T., Quinn, P. K., Sharma, S., Ström, J., and Stohl, A.: Source identification of short-lived air pollutants in the Arctic using statistical analysis of measurement data and particle dispersion model output, Atmospheric Chemistry and Physics, 10, 669–693, https://doi.org/10.5194/acp-10-669-2010, 2010b.

Ibrahim, M., Curci, G., Habbani, F. I., Kucharski, F., Tuccella, P., and Strada, S.: Association of Air Pollution Levels to Atmospheric Weather Regimes over Europe, Journal of Environmental Science and Pollution Research, 7, 442–446, 2021.

Immler, F. J., Dykema, J., Gardiner, T., Whiteman, D. N., Thorne, P. W., and Vömel, H.: Reference quality upper-air measurements: Guidance for developing GRUAN data products, Atmospheric Measurement Techniques, 3, 1217–1231, https://doi.org/10.5194/amt-3-1217-2010, 2010.

IPCC Working Group 1, I., Stocker, T., Qin, D., Plattner, G.-K., Tignor, M., Allen, S., Boschung, J., Nauels, A., Xia, Y., Bex, V., and Midgley, P.: IPCC, 2013: Climate Change 2013: The Physical Science Basis. Contribution of Working Group I to the Fifth Assessment Report of the Intergovernmental Panel on Climate Change, Tech. rep., Intergovernmental Panel on Climate Change 2013, Cambridge, United Kingdom and New York, NY, USA, http://www.ipcc.ch/pdf/assessment-report/ar5/wg1/WG1AR5_ALL_FINAL.pdf, 2013.

Johnsrud, M., Hermansen, O., and Tørnkvist, K.: Air Quality in Ny-Ålesund. Monitoring of Local Air Quality 2016-2017, Tech. rep., NILU – Norwegian Institute for Air Research, 2018.

Klima- og miljødepartementet: Lov om miljøvern på Svalbard (svalbardmiljøloven)., https://lovdata.no/dokument/NL/lov/2001-06-15-79, 2001.

Koo, J. H., Wang, Y., Kurosu, T. P., Chance, K., Rozanov, A., Richter, A., Oltmans, S. J., Thompson, a. M., Hair, J. W., Fenn, M. a., Weinheimer, a. J., Ryerson, T. B., Solberg, S., Huey, L. G., Liao, J., Dibb, J. E., Neuman, J. a., Nowak, J. B., Pierce, R. B., Natarajan, M., and Al-Saadi, J.: Characteristics of tropospheric ozone depletion events in the Arctic spring: Analysis of the ARCTAS, ARCPAC, and ARCIONS measurements and satellite BrO observations, Atmospheric Chemistry and Physics, 12, 9909–9922, https://doi.org/10.5194/acp-12-9909-2012, 2012.

Kramer, L. J., Helmig, D., Burkhart, J. F., Stohl, A., Oltmans, S., and Honrath, R. E.: Seasonal variability of atmospheric nitrogen oxides and non-methane hydrocarbons at the GEOSummit station, Greenland, Atmospheric Chemistry and Physics, 15, 6827–6849, https://doi.org/10.5194/acp-15-6827-2015, 2015.

Läderach, A. and Sodemann, H.: A revised picture of the atmospheric moisture residence time, Geophysical Research Letters, 43, 924–933, https://doi.org/10.1002/2015GL067449, 2016.

Li, J., Reiffs, A., Parchatka, U., and Fischer, H.: In situ measurements of atmospheric CO and its correlation with NOx and O3 at a rural mountain site, Metrology and measurement systems, XXII, 25–38, https://doi.org/10.1515/mms-2015-0001, 2015.

Maturilli, M.: High resolution radiosonde measurements from station Ny-Ålesund (2017-04), https://doi.org/https://doi.org/10.1594/PANGAEA.879767, 2017a.

Maturilli, M.: High resolution radiosonde measurements from station Ny-Ålesund (2017-05), https://doi.org/https://doi.org/10.1594/PANGAEA.879820, 2017b.

Maturilli, M. and Kayser, M.: Arctic warming , moisture increase and circulation changes observed in the Ny-Ålesund homogenized radiosonde record, Theoretical and Applied Climatology, 130, 1–17, https://doi.org/10.1007/s00704-016-1864-0, 2017.

Maturilli, M., Herber, A., and König-Langlo, G.: Climatology and time series of surface meteorology in Ny-Ålesund, Svalbard, Earth System Science Data, 5, 155–163, https://doi.org/10.5194/essd-5-155-2013, 2013.

Ménégoz, M., Guemas, V., Salas Y Melia, D., and Voldoire, A.: Winter interactions between aerosols and weather regimes in the North Atlantic European region, Journal of Geophysical Research Atmospheres, 115, 1–19, https://doi.org/10.1029/2009JD012480, 2010.

Monks, P. S.: Gas-phase radical chemistry in the troposphere, Chemical Society reviews, 34, 376–395, https://doi.org/10.1039/b307982c, 810 2005.

Moore, C. W., Obrist, D., Steffen, A., Staebler, R. M., Douglas, T. A., Richter, A., and Nghiem, S. V.: Convective forcing of mercury and ozone in the Arctic boundary layer induced by leads in sea ice, Nature, 506, 81–84, https://doi.org/10.1038/nature12924, 2014.

MOSJ: MOSJ (Miljøovervåking Svalbard og Jan Mayen). Antall Registrerte Snøskutere, http://www.mosj.no/no/pavirkning/ferdsel/snoskuter.html, 2018.

Papritz, L. and Grams, C. M.: Linking Low-Frequency Large-Scale Circulation Patterns to Cold Air Outbreak Formation in the Northeastern North Atlantic, Geophysical Research Letters, 45, 2542–2553, https://doi.org/10.1002/2017GL076921, 2018.

Park, S., Son, S. W., Jung, M. I., Park, J., and Park, S. S.: Evaluation of tropospheric ozone reanalyses with independent ozonesonde observations in East Asia, Geoscience Letters, 7, https://doi.org/10.1186/s40562-020-00161-9, 2020.

Parrish, D. D., Murphy, P. C., Albritton, D. L., and Fehsenfeld, F. C.: The measurement of the photodissociation rate of NO2 in the atmo-
820 sphere, Atmospheric Environment, 17, 1365–1379, https://doi.org/10.1016/0004-6981(83)90411-0, 1983.

Pasquier, J. T., Pfahl, S., and Grams, C. M.: Modulation of Atmospheric River Occurrence and Associated Precipitation Extremes in the North Atlantic Region by European Weather Regimes, Geophysical Research Letters, 46, 1014–1023, https://doi.org/10.1029/2018GL081194, 2019.

Porter, W. C., Heald, C. L., Cooley, D., and Russell, B.: Investigating the observed sensitivities of air-quality extremes to meteorological 825 drivers via quantile regression, Atmospheric Chemistry and Physics, 15, 10 349–10 366, https://doi.org/10.5194/acp-15-10349-2015, 2015.

Quinn, P. K., Bates, T. S., Baum, E., Bond, T., Burkhart, J. F., Fiore, a. M., Flanner, M. G., Garrett, T. J., Koch, D., Mcconnell, J. R., Shindell, D., and Stohl, a.: The Impact of Short-Lived Pollutants on Arctic Climate., Tech. Rep. 1, Arctic Monitoring and Assessment Programme (AMAP), Oslo, Norway, 2008.

Reimann, S., Kallenborn, R., and Schmidbauer, N.: Severe aromatic hydrocarbon pollution in the Arctic town of Longyearbyen (Svalbard) 830 caused by snowmobile emissions., Environmental science  technology, 43, 4791–4795, http://www.ncbi.nlm.nih.gov/pubmed/19673266, 2009.

Robertson, S. C., Lanchester, B. S., Galand, M., Lummerzheim, D., Stockton-Chalk, A. B., Aylward, A. D., Furniss, I., and Baumgardner, J.: First ground-based optical analysis of H$\beta$ Doppler profiles close to local noon in the cusp, Annales Geophysicae, 24, 2543–2552, https://doi.org/10.5194/angeo-24-2543-2006, 2006.

Rolph, G., Stein, A., and Stunder, B.: Real-time Environmental Applications and Display sYstem: READY, Environmental Modelling and Software, 95, 210–228, https://doi.org/10.1016/j.envsoft.2017.06.025, 2017.

Schmalwieser, A. W., Gröbner, J., Blumthaler, M., Klotz, B., De Backer, H., Bolsée, D., Werner, R., Tomsic, D., Metelka, L., Eriksen, P., Jepsen, N., Aun, M., Heikkilä, A., Duprat, T., Sandmann, H., Weiss, T., Bais, A., Toth, Z., Siani, A. M., Vaccaro, L., Diémoz, H., Grifoni,

D., Zipoli, G., Lorenzetto, G., Petkov, B. H., Di Sarra, A. G., Massen, F., Yousif, C., Aculinin, A. A., Den Outer, P., Svendby, T., Dahlback, A., Johnsen, B., Biszczuk-Jakubowska, J., Krzyscin, J., Henriques, D., Chubarova, N., Kolarž, P., Mijatovic, Z., Groselj, D., Pribullova, A., Gonzales, J. R. M., Bilbao, J., Guerrero, J. M. V., Serrano, A., Andersson, S., Vuilleumier, L., Webb, A., and O'Hagan, J.: UV Index monitoring in Europe, Photochemical and Photobiological Sciences, 16, 1349–1370, https://doi.org/10.1039/c7pp00178a, 2017.

Seinfeld, J. H. and Pandis, S. N.: Atmospheric Chemistry and Physics: From Air Pollution to Climate Change, John Wiley Sons, Inc, New York, U.S., 2nd edn., 2006.

Shears, J., Theisen, F., Bjørdal, A., and Norris, S.: Environmental impact assessment. Ny-Ålesund international scientific research and monitoring station, Svalbard, Tech. rep., Norsk Polarinstitutt, Tromsø, 1998.

Simpson, W. R., Brown, S. S., Saiz-Lopez, A., Thornton, J. A., and Von Glasow, R.: Tropospheric Halogen Chemistry: Sources, Cycling, and Impacts, Chemical Reviews, 115, 4035–4062, https://doi.org/10.1021/cr5006638, 2015.

Stein, A., Draxler, R., Rolph, G., Stunder, B., Cohen, M., and Ngan, F.: NOAA's HYSPLIT atmospheric transport and dispersion modeling system, Bulletin of the American Meteorological Society, pp. 2059–2077, https://doi.org/10.1175/BAMS-D-14-00110.1, 2015.

Stohl, A., Forster, C., Frank, A., Seibert, P., and Wotawa, G.: Technical note: The Lagrangian particle dispersion model FLEXPART version 6.2, Atmospheric Chemistry and Physics, 5, 2461–2474, https://doi.org/10.5194/acp-5-2461-2005, 2005.

Tennbakk, B., Fiksen, K., Borsche, T., Grøndahl, R., Jarstein, S., and Ramm, B.: Alternativer for framtidig energiforsyning på Svalbard, Tech. Rep. 2018-09, THEMA Consulting Group, Oslo, Norway, https://www.regjeringen.no/contentassets/cdaceb5f6b5e4fb1aa4e5e151a87859a/thema-og-multiconsult---energiforsyningen-pa-svalbard.pdf, 2018.

Trebs, I., Bohn, B., Ammann, C., Rummel, U., Blumthaler, M., Königstedt, R., Meixner, F. X., Fan, S., and Andreae, M. O.: Relationship between the NO2 photolysis frequency and the solar global irradiance, Atmospheric Measurement Techniques, 2, 725–739, https://doi.org/10.5194/amt-2-725-2009, 2009.

Vestreng, V., Kallenborn, R., and Økstad, E.: Climate influencing emissions, scenarios and mitigation options at Svalbard, Tech. rep., Klima- og forurensningsdirektoratet, Klima- og forurensningsdirektoratet, Oslo, Norway, 2009.

Wallace, J. M. and Hobbs, P. V.: Atmospheric science: an introductory survey, Academic Press, New York, 2nd edn., 2006.

Williams, E. J., Fehsenfeld, F. C., Jobson, B. T., Kuster, W. C., Goldan, P. D., Stutz, J., and McClenny, W. A.: Comparison of Ultraviolet Absorbance, Chemiluminescence, and DOAS Instruments for Ambient Ozone Monitoring, Environmental Science Technology, 40, 5755–5762, https://doi.org/10.1021/es0523542, 2006.