# Peer review of "Springtime nitrogen oxides and tropospheric ozone in Svalbard: results from the measurement station network"

_Atmospheric Chemistry and Physics, 2021_

## Referee Comment (RC1)

Comment on *Springtime nitrogen oxides and tropospheric ozone in Svalbard: results from the measurement station network* by Dekhtyareva et al.,

The paper is based on a novel set of NOx and ozone measurements from three settlements on Svalbard during spring 2017. The main objective (page 4 lines 89-90) is to: i*dentify specific factors affecting the concentration of measured compounds and define conditions that promote accumulation of local and long-range transported pollution in all three settlements.*.

The paper do address relevant scientific questions within the scope of ACP, and it does present novel data from the measurements on Svalbard and applies a semi-novel concept linking atmospheric circulation regimes and pollution levels.

Although several possible environmental impacts are mentioned in the introduction, objective/science questions on page 4 and the discussion/conclusion sections it becomes clear that the primary objective is to study the air quality (wrt NOx and ozone) in the three settlements and how this is related to atmospheric conditions.

The main issue with the paper is the inconsistency of the data. This relates to the lack of co-location for the NOx and ozone measurements, no ozone measuements in Adventdalen, and the issues with calibration of the instruments in Barentsburg (as also pointed out the authors at the end of the discussion section). This has effects on the conclusions that can be drawn, in that these can not be very strong. The manuscript includes in several places statements like "This *may be explained* by the fact that the measurement station in Ny-Ålesund was located much closer to the diesel power plant, ….. ". Also, the discussion linking circulation regimes and air quality lacks clarity and could benefit from including more years (using ozone data from Zeppelin) to get a more robust result.

I find that the description and discussion in many places could have been more precise (cf detailed comments below).

I suggest that the manuscript can be published after major revisions.

Specific Comments:

Page 2, line 43. There is a very appropriate reference to Beine et al., 1997 who found that PAN decomposition was an important source of NOx at the Zeppelin station. However, this point has not been taken up again during the discussions.

Page 3 line 68: Is it really true that there is no diurnal cycle in the demand for electricity in Longyearbyen? What about the use for cooking and light with a peak in the afternoon?

Page 7 lines 160-170. It is unclear what this "no regime" is. The "for example" wording on line 160 is very confusing.

Page 7 lines 167-171. The sentence starts with "Secondly". Does this means that the identification of the flow regimes for the sub-periods was done on another dataset than the ERA5?

Page 10 line 248: The amplitudes are not very high (+/- 2ppb or so), but they are very short term fluctuations.

Page 11 line 268: Define ozone titration efficiency

Page 11 line 267-271: First, it is stated that there is a strong negative correlation, and then later (line 270) it is stated that it is not statistically significant. This seems contradictory.

Page 11 line 281. The word "However" seems misplaced here. Since the measurements are so close to the source this kind of extreme values can be expected there.

Page 10 line 247. It is concluded that the synoptic conditions have minor effects on NOx due to the low correlation between Ny-Ålesund and Barentsbug. However, from the maps (figure 6) it is clear that in Ny-Ålesund the source is North of the station, while it is the opposite in Barentsburg. Thus, I would expect that the wind direction component of the synoptic conditions could give negative correlation.

On the NO/NOx relation

Generally there will be a local steady state given by the reactions

NO + O3 ➔ NO2 + O2          R1

NO2 + hv (+O2) ➔ NO + O3      R2

At steady state

[NO] = [NO2]* k2/(k1*[O3]

The paper seems to neglect the effect of reaction R2 for the NO2/NOx ratio. With appropriate values for k1, [O3] and k2 (the photolysis rate) one can derive the steady state NO2/NOx ratio. It would be of interest to see how the ratios observed in Adventdalen during daytime is affected by the photolysis.

Figure 4. The diurnal cycle of the means of ozone at BB and Zeppelin are shown. These are the means over the springtime period (about 55 days) I presume. Please add to this figure the standard error of the mean for each hour, so that we can see if these diurnal cycles are really statistically significant.

Page 12, line 305. I don't understand the argument that enhanced photolysis is compensated by convection. Photolysis in itself would not reduce ozone significantly as the reaction O + O2 + M ➔ O3 + M would very rapidly reform ozone. In addition, I would expect that for Zeppelin convection would mix in PBL air with lower ozone. At very low NOx levels there could be enhanced loss of ozone during daytime through O3 + OH ➔ O2 + HO2 followed by O3 + HO2 ➔ OH + 2 O2

Page 12, line 307: If the diurnal cycle is not statistically significant, there is no need to discuss possible physical/chemical explanations!

Page 12 line 312: Wind speed 4.1 m/s. This must be the average wind speed. Please also give the variance.

Page 13 line 317:  You have written: normally ventilation is sufficient to remove $NO_x$ emitted by the usual amount of motorized traffic. I don't understand this statement: is NOx completely removed?

Page 15 line 342: Unclear sentence. Is the 46% referring to the whole period or sub-period I? Is 0.95 °C the median or is it the deviation from the median during sub-period  I.

Figure 7. It would be useful to have the sub-periods indicated over the individual plots.

Page 14, Section 3.2: I find this whole section quite unorganized. The whole section seems to focus on PBL high and how it affects NOx, and possible transport patterns for ozone that allow ozone depletion events. There is a lack of motivation for selecting these regimes. E.g. the regimes depicted in fig 7a and 7h looks very similar to me, and without  a rational for splitting this in two different regimes. A factor that is completely missing is the possibility of tropopause folding events with intrusion of ozone rich air, presumable related to the circulation regimes.

On the transport of pollutants to Svalbard the work by Hirdman et al. should be referenced (Hirdman et al., Atmos. Chem. Phys., 10, 9351–9368, 2010 www.atmos-chem-phys.net/10/9351/2010/ doi:10.5194/acp-10-9351-2010)

Page 16, line 366-380. Elevated NOx concentrations were found in Adventdalen but not in Barentsburg for period VI with cold conditions and low PBL height. These condition with low wind (and maybe clear sky) would I believe enhance recreational snow mobile traffic and thus emissions, which is much more pronounced in Longyearbyen than in Barentsburg. In general, there may be a link between weather and emissions that is not mentioned in the paper.

Page 18, line 382-390. The paper concludes (elsewhere) that there is high correlation between ozone at Zeppelin and Barentsburg, thus the measurements represent region ozone levels. Since ozone data from Zeppelin is available for a number of years, this regime analysis could be extended for a much longer period and thus be much more robust.

Figure 8:

-This is not really the trajectory probability for the different regimes, but rather for the sub-periods. Having a longer (multi year) record to make these probabilities for the regimes would help. Very difficult to read the red contours.

- The maps are very small. There is no need to include the same label bar 9 times. Also I recommend that each map is labeled with the name of the corresponding circulation regime (applies also to figure 7).

Page 19, line 396. Why is the HYSPLIt model used for these back trajectories and not Flexpart?

Page 22, line 474. The authors claim that "The weather regime approach is novel in the air pollution research". However, this has been used in several studies before, although not for the Svalbard region I believe. See references below.

Muntasir A. Ibrahim*, Gabriele Curci, Farouk I. Habbani, Fred Kucharski, Paolo Tuccella, Susanna Strada, Linking weather regimes and air pollution/air quality.

https://doi.org/10.30799/jespr.210.21070101

M. Ménégoz,V. Guemas,D. Salas y Melia,A. Voldoire, https://doi.org/10.1029/2009JD012480

---

## Referee Comment (RC2)

This paper reports springtime nitrogen oxides and tropospheric ozone measurements in Svalbard, using observations on 3 stations, during less almost 2 months in spring 2017, and Lagrangian backward trajectory analysis.

The observations itself are of interest because in the Svalbard are really sparse, so they can help to understand formation and evolution of $O_3$ in a remote area effected by interesting and mixed effects such us depletion due to reactive halogen compounds and photochemistry due to local well defined emission sources. However, there are several shortcomings that limit a lot the results of this paper, therefore I'm really skeptical to suggest to accept this manuscript for publication on ACP, unless it will be deeply revised, for the following reasons:

1) Only in the Barentsburg Station are observed both $O_3$ and NOx, that are the two species fundamental for this study, in the Adventdalen and Ny-Alesund sites are missed the measure of O3, therefore a comparison and correlations of these species among these station is misleading. To partially overcome this problem for the Ny-Alesund analysis are used the $O_3$ measurements of the Zeppelin observatory, but this is not the solution not only because the latter is 2 km away from Ny-Alesund site, but, more seriously, it is on the top of a mountain at 474m. a.s.l., whereas Ny-Alesund site is at 23 m. a.s.l. and near the sea. Finally, since in Ny-Alesund site are missed also meteorological measurements, for the analysis on this site are used data of the top of the mountain (Zeppelin station). It would have been much more worth to install the $NO_x$ analyser in the Zeppelin station where meteorological data and $O_3$ measurements were available then in the Ny-Alesund site.

2) Lines 233-236: Since the CO measurements were stable at Zeppelin station so no sharp peaks to identify local emission were detected, this is a proof that the measurements at Ny-Alesund are not useful to understand potential impact of local pollution on $O_3$ evolution in the mountain top.

3) Lines 243-247: Here the Author affirm that local emissions are important because the correlation between $NO_x$ measurements at Ny-Alesund and Adventdalen sites are weak, but at lines 233-236, looking at CO data they assert that local pollution are not important for the area under investigation. This is contradictory conclusion is due to another big issue: since CO measurements are available only at Zeppelin station, the signature of local emissions in other sites (where CO were not measured), were tried to find in the correlation between $NO_x$ observation, again measurements of CO in Ny-Alesund and Adventdalen site would have been worth to make this conclusion.

4) Lines 255-257: Author here assert that synoptic transport is more important than local emission looking, now, at the correlation of $O_3$ measurements at Zeppelin and Barentsburg site, a couple of issues: a) r = 0.69 is considered a 'strong correlation', it means $r^2$ = 0.47, that is not that 'strong', b) again since $O_3$ is missed in the Ny-Alesund and Adventdalen, now to decide if dominate local emission or synoptic transport the correlation od $O_3$ between Zeppelin and Barentsburg site are used, while before (lines 233-236) were used $NO_x$ for Ny-Alesund and Adventdalen sites, obtaining contradictory results.

5) Lines 295-298 and figure 4: The $O_3$ diurnal cycle of Barentsburg site is, as expected, completely different of that of the Zeppelin station, in the first is evident the typical profile dominated by photochemistry, in the later, a typical mountain station data, with no diurnal cycle. This is what expected, but again in contrast with what reported in lines 255-257 where Author affirm that the $O_3$ measured in these two sites showed a 'strong correlation': they have a completely different dynamics, as can be expected since one is in a mountain and the other in a site at 40 m. a.s.l.

6) Lines 385-399: Here initially, looking at trajectory analysis Authors affirm that the $O_3$ decrease can be explained by local depletion due to air masses rich of BrO, whereas at the end is supposed that may be due to less photochemistry due to 'lack of sunlight and $O_3$ precursors such as $NO_x$'. A good result of this analysis would have been if the two effects (depletion due to BrO vs photochemistry and $NO_x$ emission) were well characterized and, from observations and model

analysis, quantified and compared, unfortunately here both effects are claimed, as can be guessed even without any kind of measurements and/or model analysis.

7) Lines 408-411: Finally, the Authors, looking $O_3$ sondes data confirm that Zepellin data are different of that in Barentsburg because one is at the top of Mountain and the other at 40 m. a.s.l. I think that this would have been the first analysis of the paper, and not the last one after different correlation analysis where was mentioned that $O_3$ data of those sites are strongly correlated.

8) Lines 471-473: From the data and analysis I'm not comfortable with the conclusion that local emission of $NO_x$ may reduce $O_3$ level by few percent in the Ny-Alensund site, since $O_3$ there is not measured, but, again, are used for this conclusion $O_3$ measured at the top of mountain. Here for example, since $O_3$ where not measured would have been worth to use a Box model (such as MCM) to model $O_3$ at Ny-Alensund, constrained by local NOx measurements.

9) Lines: 484-486: From data analysis and model simulation it is hard to support these conclusions.

A general comment and a suggestion for further observations in this area: $NO_2$ measurements in remote area, where the concentrations are very low may have bias or instruments are below the detection limits for most of the times. There are several papers that suggest to use instrument that measure directly $NO_2$, using CAPS, LIF or CRDS techniques (actually CAPS are commercially available now) or, at least, chemiluminescence systems that uses photolytic conversion of $NO_2$ into NO, besides systems like those used in this work (model T200) that uses molybdenum oxide converters (Steinbacher et al., 2007; Dunlea, et al. 2007; Yang et al., 2004; Villena et al., 2012).

Minor comment:

Line 391: When in a time period, 67% of data are missed analysis and conclusion are very weak, so may be better not include that period in the analysis.

Reference
Steinbacher, M., Zellweger, B. Schwarzenbach, S. Bugmann, B. Buchmann, C. Ordonez, A. S. H. Prevot, and C. Hueglin, Nitrogen oxide measurements at rural sites in Switzerland:Bias of conventional measurement techniques, Journal of Geophy. Research, 112, D11307, doi:10.1029/2006JD007971, 2007.
Dunlea, E. J., Herndon, S. C., Nelson, D. D., Volkamer, R. M., San Martini, F., Sheehy, P. M., Zahniser, M. S., Shorter, J. H., Wormhoudt, J. C., Lamb, B. K., Allwine, E. J., Gaffney, J. S., Marley, N. A., Grutter, M., Marquez, C., Blanco, S., Cardenas, B., Retama, A., Ramos Villegas, C. R., Kolb, C. E., Molina, L. T., and Molina, M. J.: Evaluation of nitrogen dioxide chemiluminescence monitors in a polluted urban environment, Atmos. Chem. Phys., 7, 2691–2704, doi:10.5194/acp-7-2691-2007, 2007.
Yang, J., Honrath, R. E., Peterson, M. C., Parrish, D. D., and Warshawsky, M.: Photostationary state deviation–estimated peroxy radicals and their implications for HOx and ozone photochemistry at a remote northern Atlantic coastal site, J. Geophys. Res., 109, D02312, doi:10.1029/2003JD003983, 2004.
Villena, G., Bejan, I., Kurtenbach, R.,Wiesen, P., and Kleffmann, J.: Interferences of commercial NO2 instruments in the urban atmosphere and in a smog chamber, Atmos. Meas. Tech., 5, 149–159, doi:10.5194/amt-5-149-2012, 2012.

---

## Author Response (AR1)

We would like to thank Reviewers 1 and 2 for the thorough revision of the manuscript and insightful comments that allowed to improve the quality of the paper and make more solid conclusions.

In addition to corrections performed according to comments from the Reviewer 1 and 2, the following major changes in the manuscript have been introduced:
1) Following suggestion of the Reviewer 2, the description of $O_3$ profiles has been moved to the beginning of the Results section and Figure 10 became Figure 4 in the new manuscript, and numbering of the other figures has been modified accordingly.
2) In the Discussion (section 4), the results from 2017 campaign are contrasted to modelled $NO_2/NO_x$ ratio and $NO_2$ produced through PAN decomposition, and long-term observations from Ny-Ålesund, weather regime and trajectory data are utilized to confirm weather regime and air pollution links.
3) Four new plots have been added in the Appendix and discussed in the Discussion section.
4) The Conclusion section is revised to reflect the changes introduced in the manuscript.

A point-by-point response to the reviews in given below with Reviewers' comments shown in cursive and response shown in normal font.

**1. Reply to the Reviewer 1**

*The paper is based on a novel set of NOx and ozone measurements from three settlements on Svalbard during spring 2017. The main objective (page 4 lines 89-90) is to: identify specific factors affecting the concentration of measured compounds and define conditions that promote accumulation of local and long-range transported pollution in all three settlements..*
*The paper do address relevant scientific questions within the scope of ACP, and it does present novel data from the measurements on Svalbard and applies a semi-novel concept linking atmospheric circulation regimes and pollution levels.*
*Although several possible environmental impacts are mentioned in the introduction, objective/science questions on page 4 and the discussion/conclusion sections it becomes clear that the primary objective is to study the air quality (wrt NOx and ozone) in the three settlements and how this is related to atmospheric conditions.*
*The main issue with the paper is the inconsistency of the data. This relates to the lack of co-location for the NOx and ozone measurements, no ozone measuements in Adventdalen, and the issues with calibration of the instruments in Barentsburg (as also pointed out the authors at the end of the discussion section). This has effects on the conclusions that can be drawn, in that these can not be very strong. The manuscript includes in several places statements like "This may be explained by the fact that the measurement station in Ny-Ålesund was located much closer to the diesel power plant, ….. ". Also, the discussion linking circulation regimes and air quality lacks clarity and could benefit from including more years (using ozone data from Zeppelin) to get a more robust result.*
*I find that the description and discussion in many places could have been more precise (cf detailed comments below).*
*I suggest that the manuscript can be published after major revisions.*

**Reply to the major comments**

We agree that the lack of $NO_x/O_3$ instrument co-location in Adventdalen and Ny-Ålesund during the 2017 campaign is a major drawback of this study. The $NO_x$ monitoring in Ny-Ålesund is a long-term ongoing air quality project, and relocation of the instrument from the village to the Zeppelin station would introduce bias in the long-term observations in the settlement. The study in Adventdalen was the

first combined air pollution and meteorological field work in Longyearbyen. The measurements there were done by the main author, and only $NO_x$ monitor was installed there due to the limited grant funding.

Regarding $NO_x$ and $O_3$ data from Ny-Ålesund, the authors have been in contact with atmospheric scientists from Ny-Ålesund research community, and, to our knowledge, the only collocated $NO_x$ and $O_3$ measurements from this settlement were performed at the Zeppelin station from February to May 1994. The results were published in Beine et al., 1996. In that study, the combination of $NO_x$ data and concentration of particles with diameter below 10 nm, atmospheric stability and wind direction was used to identify possible local pollution events. In spring 1994, the local pollution was detected at the Zeppelin station during 6.4 % of the measurement time, and the number of these events was increasing with increased insolation in May. Following this method of event detection, the concentration of particles with diameter of 10 nm routinely measured by DMPS at the Zeppelin station and threshold of > 95 percentile have been used to identify peaks in concentration of newly formed particles. Similarly to the results of Beine et al., 1996, the peak events were detected at the Zeppelin station only in the second part of the 2017 measurement campaign (from $24^{th}$ of April to $13^{th}$ of May). The northerly wind direction was present only during 12 out of 45 hours with peak particle concentration, however, none of these cases was characterized by increased CO concentration at the Zeppelin station. Thus, these peaks in concentration of small particles might have been related to natural rather than anthropogenic emission sources. Indeed, Heintzenberg et al., 2017 described the offset in new particle formation towards late spring and summer when biological emissions become important sources for this process. Therefore, both statistical comparison of the $O_3$ concentrations in clean and potentially polluted air masses mentioned in the manuscript and absence of coinciding peaks in particle concentration and CO concentration indicate that the $O_3$ observations at the Zeppelin station were not significantly affected by local $NO_x$ pollution during 2017 campaign. Therefore, the lines 226-244 in the section 2.5 have been rewritten and the results about influence of local $NO_x$ pollution on the $O_3$ concentrations at the Zeppelin station have been modified accordingly (lines 288-305 in the new version of the manuscript).

To investigate the influence of local $NO_x$ emissions on the $O_3$ concentration in Ny-Ålesund, the following part has been included in the Discussion part of the paper (lines 529-543).

To investigate the influence of local $NO_x$ emissions on the $O_3$ concentration in Ny-Ålesund, as it is required in the third hypothesis stated in the introduction of the current paper, we may use historical observations. The data from only six $O_3$ sonde launches were available for the 2017 campaign (Figure 4). However, the long-term data below 100 m from the $O_3$ sonde profiles may be used to study influence of the local $NO_x$ pollution in Ny-Ålesund on the $O_3$ concentration. These observations are suitable for this purpose because the $O_3$ sonde launching facility is located just 200 m to the south-south-west and 500 m to the south from the $NO_x$ monitor and diesel power plant, respectively. Thus, when the monitor detected $NO_x$ concentration above long-term springtime average in the launch hour, the influence of locally polluted air masses might have been observed in the lowest $O_3$ sonde data. There were in total 59 $O_3$ sonde launches, for which $NO_x$ monitor data was available in spring 2009, 2010, 2015, 2016, 2017 and 2018. The $O_3$ profile data in the lowest 100 m have been extracted for all 59 launches and grouped according to the $NO_x$ concentration detected by the monitor and wind direction in the $O_3$ sonde profiles: 1) above mean $NO_x$ concentration and northerly wind direction; 2) below or equal to mean $NO_x$ concentration and northerly wind direction. The median and mean $O_3$ values below 100 m in the group where the $NO_x$ values were above $NO_x$ mean were 11 % and 15% lower, respectively, than for the second group with northerly winds, but without elevated $NO_x$ concentration. Thus, the $O_3$ concentration in lowest 100 m downwind from the power plant in the settlement may be reduced significantly due to local $NO_x$ emissions, but the frequency of such events is unknown in absence of continuous $O_3$ measurements in the village.

Following the advice of the Reviewer 1, we included analysis of more years into the discussion part to get a more robust result linking weather regimes and air quality (lines 567-611).

To get a more robust result linking weather regimes and air quality, we would like to compare long-term springtime (23 March-15 May) weather regime data with $NO_x$ data from Ny-Ålesund, $O_3$ concentration from the Zeppelin station, $O_3$ sonde and radiosonde data as well as FLEXPART trajectories for a period from 1990 to 2018. The FLEXPART and weather regime data were available for all years, while there were gaps in observational data from Ny-Ålesund. The data availability chart is shown in Figure A3 indicating number of hourly measurements for surface $NO_x$ and $O_3$ data and number of radiosonde and ozone sonde launches per spring season each year. The hourly $O_3$ data is available for all years, while hourly $NO_x$ data was only available in 2009, 2010, 2015, 2016, 2017 and 2018. After spring 2018, the $NO_x$ monitor was moved to other location in Ny-Ålesund, therefore 2019 - 2022 data are not included in the current analysis to keep measurement consistency. The $O_3$ soundings and radiosonde AWI's datasets start in 1992 and 1993 with median number of radiosonde soundings and $O_3$ soundings per spring season being 54 and 11, respectively.

[revised manuscript text omitted]

**Reply to specific comments**

*Page 2, line 43. There is a very appropriate reference to Beine et al., 1997 who found that PAN decomposition was an important source of $NO_x$ at the Zeppelin station. However, this point has not been taken up again during the discussions.*

We would like to thank the Reviewer 1 for the useful comment. We have included the following part on PAN decomposition into the discussion (lines 501-522):

The results from radiosonde and ozone soundings, CO and particle measurements, presented in this study, demonstrate that the $O_3$ observations at the Zeppelin station were not sensitive to the local $NO_x$ pollution from Ny-Ålesund, and thus were representative as background values for comparison with Barentsburg and investigation of the influence of prevailing long-range transport patterns on the measurements at these stations. Furthermore, $O_3$ data from the Zeppelin station may be used to assess how the PAN decomposition might have affected the background $NO_x$ concentrations in Svalbard during the 2017 campaign.

Previous studies have shown that the $NO_x$/PAN ratio increases at temperatures above -10 $^o$C, and PAN decomposition becomes a major source of background $NO_x$ in Svalbard (Beine et al., 1997a; Beine and Krognes, 2000). As the temperature at the Zeppelin station varied from -22.7 to 0.8 $^o$C during the campaign, we would like to investigate the contribution of PAN decomposition to the background $NO_x$ concentration in Svalbard. The PAN decomposition rate may be estimated using several approaches (Beine et al., 1997a), here we apply a linear relationship between $O_3$ and PAN concentration derived from previous measurements at the Zeppelin station: PAN[ppt]=($O_3$[ppb]-26.58)/0.034 and then calculate PAN decomposition rate (Beine and Krognes, 2000) (Figure A2a). The maximum PAN decomposition rate has been calculated using temperatures and $O_3$ concentration observed at the Zeppelin station applying equation (1) from Beine and Krognes, 2000 (Beine and Krognes, 2000) . The depletion events when $O_3$ concentration was below 26.58 ppb have been excluded from the calculation (Beine and Krognes, 2000). The median calculated PAN concentration of 356 pptv (ppb x$10^{-3}$ ) is comparable with previous springtime Arctic observations (Beine and Krognes, 2000; Kramer, 2015) .The estimated maximum PAN decomposition rate for the whole campaign varied from -0.0033 to -17.2 pptv hour$^{-1}$ with median value of -1.29 pptv hour$^{-1}$ (Figure A2b). The maximum PAN concentration coincides with the strongest $O_3$ increase event occurred 03.05.2017 (Figure 10b). The temperature increased simultaneously for that day (Figure 2a) promoting efficient PAN decomposition (Figure A2b). Applying Theil's non-parametric regression with the slope of –5.07 (pptv $NO_x$/pptv hour$^{-1}$ PAN) suggested by Beine et al., (1997a) for Svalbard, the background concentration of $NO_x$ would be 87.2 pptv. However, these concentrations are too low for the equipment used in the 2017 campaign to detect the variations in the concentrations caused by the PAN decomposition.

*Page 3 line 68: Is it really true that there is no diurnal cycle in the demand for electricity in Longyearbyen? What about the use for cooking and light with a peak in the afternoon?*

We would like to thank reviewer for the comment. The sentences have been corrected as following (line 68-71):
In contrast to the energy needed for heating, the energy demand for electricity production is mostly independent on the air temperature. Industry, business and communal buildings stand for more than 70% of the electricity consumption in Longyearbyen. There is a diurnal variation in the power demand with higher daytime values in winter.

*Page 7 lines 160-170. It is unclear what this "no regime" is. The "for example" wording on line 160 is very confusing.*

The sentence has been changed as following (lines 180-182):
In the climatological mean, the large-scale conditions are characterized by weak ridging of absolute geopotential height at 500 hPa over the eastern North Atlantic and westerly upper level flow over Svalbard. Such regime is placed in the "no regime" category in the Grams et al., 2017 classification (their Fig. S1h).

*Page 7 lines 167-171. The sentence starts with "Secondly". Does this means that the identification of the flow regimes for the sub-periods was done on another dataset than the ERA5?*
Following sentence has been added into the manuscript: (line 172).
In the current work, we apply Dr. Christian Grams's weather regime classification that is based on the 6-hourly ERA-Interim data.

*Page 10 line 248: The amplitudes are not very high (+/- 2ppb or so), but they are very short term fluctuations.*
The sentence has been corrected as following (lines 277-278):
The Barentsburg $O_3$ data contains some abrupt peaks with magnitude of up to 9 ppbv and duration of just one hour (light blue line in Figure 3c)), while they are absent in the Zeppelin $O_3$ data.

*Page 11 line 268: Define ozone titration efficiency.*
The sentence has been change as following (lines 306-308):
Difference between the original and smoothed $O_3$ data from Barentsburg varies from -19% to 11% of the smoothed value, and there is a moderate negative correlation between the magnitude of $NO_x$ peak and reduction in $O_3$ concentration (r=-0.65, p<0.0001).

*Page 11 line 267-271: First, it is stated that there is a strong negative correlation, and then later (line 270) it is stated that it is not statistically significant. This seems contradictory.*
The sentence has been rewritten as following (lines 308-311):
Despite this sensitivity of $O_3$ concentration to local pollution in Barentsburg, the median $NO_x$ concentrations observed there were low and average reduction of $O_3$ concentration in comparison to the smoothed values was only 1%. This effect is not statistically significant, and therefore other factors, such as variation in concentrations within long-range transported air masses, masses, may be more important for explanation of difference between the $O_3$ Zeppelin and Barentsburg datasets.

*Page 11 line 281. The word "However" seems misplaced here. Since the measurements are so close to the source this kind of extreme values can be expected there.*

The word "However" has been deleted.

*Page 10 line 247. It is concluded that the synoptic conditions have minor effects on $NO_x$ due to the low correlation between Ny-Ålesund and Barentsbug. However, from the maps (figure 6) it is clear that in Ny-Ålesund the source is North of the station, while it is the opposite in Barentsburg. Thus, I would expect that the wind direction component of the synoptic conditions could give negative correlation.*

We would like to thank the Reviewer for the comment. Indeed, one may expect negative correlation for the stations, but the atmospheric stability also plays role, since the monitoring station in Barentsburg is located on the hill above the pollution source, while stations in Ny-Ålesund and Adventdalen are located on the same level or below emission sources. The combination of these factors caused by the synoptic situation affect the correlation of the measurement results from the three stations. The sentence has been changed accordingly (lines 271-276):

On the contrary, no correlation is present with $NO_x$ data from Barentsburg. Low correlation between the $NO_x$ values at the three stations indicates the importance of local emission sources and micrometeorology (wind channelling and spatial variation in atmospheric stability) rather than synoptic meteorological conditions. The background $NO_x$ concentrations observed in Svalbard in previous studies (Beine et al., 1997) using different measurement techniques are below 0.4 ppb, and thus, the natural variability in $NO_x$ values due to long-range transport to Svalbard would be undetected in the $NO_x$ datasets presented in the current study.

*The paper seems to neglect the effect of reaction R2 for the $NO_2/NO_x$ ratio. With appropriate values for k1, $[O_3]$ and k2 (the photolysis rate) one can derive the steady state $NO_2/NO_x$ ratio. It would be of interest to see how the ratios observed in Adventdalen during daytime is affected by the photolysis.*

We would like to thank for this comment. The section about the effect of photolysis is added in the discussion (lines 474-501):
The $NO_x$ monitor in Adventdalen was located far away from stationary emission sources and showed the highest daytime $NO_2/NO_x$ ratio (Table 1). We would like to investigate how the ratios observed there were affected by photolysis. The photolysis rate of $NO_2$ depends on solar zenith angle (Parrish et al. , 1983), which in turn depends on day of year. Measurements were performed between days 81 and 134, and the noon solar zenith angle in Longyearbyen area varied from approximately 77° to 62° (Robertson et al. , 2006). Following equation (15) in Parrish et al. (1983), minimum clear-sky photolysis rate for the start of the campaign was 0.0026 $s^{-1}$, and maximum clear-sky photolysis rate for the end of the campaign was 0.0061 $s^{-1}$ (black squares in Figure A1a)). There are many factors that affect $NO_2$ photolysis rate, such as aerosol load, clouds, water vapour content and surface albedo (Trebs et al. , 2009). The albedo may significantly increase the $NO_2$ photolysis rate (Trebs et al. , 2009), and Dickerson et al. (1982) suggested albedo of snow with respect to $j(NO_2)$ to be 93%. Trebs et al. (2009) suggested in their equation (2) a polynomial fit between global irradiance and $NO_2$ photolysis rate that includes both clear-sky and cloudy conditions and takes into account the contribution of albedo. The albedo calculated as the ratio of upward and downward short-wave radiation measured by CNR1 Kipp Zonen net radiometer in Adventdalen and observed global radiation were used to estimate $j(NO_2)$ (red line in Figure A1a)). Figure A1b) shows $NO/NO_2$ ratio calculated using $O_3$ concentration measured in Barentsburg (closest station where $O_3$ measurements were available), $j(NO_2)$ and temperature-dependent rate coefficient $kNO+O_3$ obtained using temperatures in Adventdalen (equation 6.6 and Table 6.1 in Seinfeld Pandis, 2006). The peaks of $NO/NO_2$ ratios are especially pronounced for the days with decreased $O_3$ concentration (01.04.2017 and the period 485 from 04.05.2017 to 09.05.2017). Note, the calculation is based on the $O_3$ data from Barentsburg, thus this introduces an uncertainty in the exact $NO/NO_2$ ratios estimated for Adventdalen. The observed and calculated $NO_2/NO_x$ ratio for Advendalen are shown in Figure A1c). The missing data in the observed $NO_2/NO_x$ ratio (blue line) indicate that

both NO and NO$_2$ values were within zero-noise level, while missing data in the calculated NO$_2$/NO$_x$ ratio is due to missing O$_3$ observations in Barentsburg. The observed and calculated values are of the same order, but NO$_2$/NO$_x$ ratio is underestimated in 64% of all available data, especially for the days with low O$_3$ values. This underestimation was present even in hours influenced by fresh local NO emission (green line) and might have resulted from the modelling errors that could occur if the surface albedo was high (Trebs et al. , 2009) or because the actual O$_3$ values in Adventdalen were lower than in Barentsburg. The NO$_2$/NO$_x$ ratio is overestimated in 31% of all available data. In these hours, the actual O$_3$ concentration might have been higher in Adventdalen than in Barentsburg (used for calculations). The most pronounced overestimation is noticeable in the period from 26.04 to 29.04 when NO values in Barentsburg were higher than in Adventdalen, and thus more pronounced O$_3$ titration with local NO might have occurred in Barentsburg.

*Figure 4. The diurnal cycle of the means of ozone at BB and Zeppelin are shown. These are the means over the springtime period (about 55 days) I presume. Please add to this figure the standard error of the mean for each hour, so that we can see if these diurnal cycles are really statistically significant.*

The standard error of the mean has been included for each hour (Figure 5 in the new manuscript). To improve readability of the figure, separate subplots a) and b) have been made for NO$_x$ and O$_3$ data, and supporting text has been modified accordingly.

*Page 12, line 305. I don't understand the argument that enhanced photolysis is compensated by convection. Photolysis in itself would not reduce ozone significantly as the reaction $O + O_2 + M \rightarrow O_3 + M$ would very rapidly reform ozone. In addition, I would expect that for Zeppelin convection would mix in PBL air with lower ozone. At very low NO$_x$ levels there could be enhanced loss of ozone during daytime through $O_3 + OH \rightarrow O_2 + HO_2$ followed by $O_3 + HO_2 \rightarrow OH + 2 O_2$*
*Page 12, line 307: If the diurnal cycle is not statistically significant, there is no need to discuss possible physical/chemical explanations!*
The sentences have been modified as following (lines 359-364):
In contrast, the Zeppelin station is located at the altitude of 474 m a.s.l. and mostly samples air from the free troposphere with higher O$_3$ concentration, and thus the data from this station does not exhibit similar diurnal variation as the Barentsburg station. However, the magnitude of these effects is small, and according to the t-test and the WRS-test, there is no statistically significant difference between the nighttime and daytime O$_3$ concentrations measured at the stations (Table 1).

*Page 12 line 312: Wind speed 4.1 m/s. This must be the average wind speed. Please also give the variance.*
The standard deviation of the mean has been included in the sentence (line 365).

*Page 13 line 317: You have written: normally ventilation is sufficient to remove NO$_x$ emitted by the usual amount of motorized traffic. I don't understand this statement: is NO$_x$ completely removed?*
The sentence has been modified as following (lines 375-378):
Such low wind speed is untypical for the wind regime in Adventdalen, where normally ventilation is sufficient to effectively disperse NO$_x$ emitted by the usual amount of motorized traffic.

*Page 15 line 342: Unclear sentence. Is the 46% referring to the whole period or sub-period I? Is 0.95 °C the median or is it the deviation from the median during sub-period I.*
The sentence has been corrected in the following way (lines 401-402):
In the sub-period I, the temperature inversions were observed in 46% of the radiosonde profiles from Ny-Ålesund, but the median inversion strength was below 0.95ºC (median for the whole campaign).

*Figure 7. It would be useful to have the sub-periods indicated over the individual plots.*
The figure has been modified as suggested by the Reviewer (Figure 8 in the new manuscript).

*Page 14, Section 3.2: I find this whole section quite unorganized. The whole section seems to focus on PBL high and how it affects $NO_x$, and possible transport patterns for ozone that allow ozone depletion events. There is a lack of motivation for selecting these regimes. E.g. the regimes depicted in fig 7a and 7h looks very similar to me, and without a rational for splitting this in two different regimes. A factor that is completely missing is the possibility of tropopause folding events with intrusion of ozone rich air, presumable related to the circulation regimes.*

We would like to thank the Reviewer for the comment. The weather regimes were identified as described in Section 2.4 based on the prevailing large-scale atmospheric circulation patterns and 500 hPa geopotential height. This classification was done as described in Grams et al., 2017, independently on the in-situ meteorological or chemical data. The same large-scale regimes occurred in different sub-periods throughout the campaign were analysed separately because the sea-ice conditions, snow cover, insolation, temperatures and boundary layer height evolve as seasonal transition takes place from March to May in Svalbard. Thus, even if large scale transport patterns may look similar, the concentrations of measured compounds may be different due to local processes.
The downward transport from the free troposphere is, indeed, a significant source of lower-altitude $O_3$ in Svalbard region, particularly during winter and spring (Hirdman et al., 2009). Thus, not only low-level trajectories need to be taken into account, but also percentage of trajectories descending from upper levels (>2000 m) during different sub-periods.
The respective discussion the height of transported air masses for different periods obtained in FLEXPART and HYSPLIT trajectories and importance of trajectory height for the $O_3$ concentration has been included in lines 451-459:
The $O_3$ concentration in Barentsburg was above median for this sub-period, and the trajectory data show the air masses arriving from the south-east (Figure 9b). As in previous studies of Hirdman et al. (2009), the downward transport of $O_3$-enriched air masses from higher altitudes played significant role during the 2017 campaign. The percentage of trajectory points reaching elevations above 2000 m was highest for the sub-periods III, VI and IX (27%, 33% and 24% of the total number trajectory points for each sub-period respectively). In contrast, during the sub-period VIII with the lowest $O_3$ concentration at both stations, the percentage of elevated trajectory points was minimal, only 4%. One can also see that the percentage of elevated trajectories varies for the same type of weather regime and determines importance of the downward air mass transport for the measured surface $O_3$ concentrations in different sub-periods (e.g. ScTr regime in Figure 9c) and e) and Table2).

*On the transport of pollutants to Svalbard the work by Hirdman et al. should be referenced (Hirdman et al., Atmos. Chem. Phys., 10, 9351–9368, 2010 www.atmos-chem-phys.net/10/9351/2010/ doi:10.5194/acp-10-9351-2010)*
The reference has been included in the lines 192-193 as well as two additional relevant references on FLEXPART and Arctic long-range transported pollution (Hirdman et al., 2009; Hirdman, Burkhart, et al., 2010; Hirdman, Sodemann, et al., 2010).

*Page 16, line 366-380. Elevated $NO_x$ concentrations were found in Adventdalen but not in Barentsburg for period VI with cold conditions and low PBL height. These condition with low wind (and maybe clear sky) would I believe enhance recreational snow mobile traffic and thus emissions, which is much more pronounced in Longyearbyen than in Barentsburg. In general, there may be a link between weather and emissions that is not mentioned in the paper.*

The observations of number of snowmobiles during the campaign were scarce, but the following might be stated (lines 367-373):

To test if the size of snowmobile motorcade has an influence on the $NO_x$ concentration in Advendalen, manual observations of number of snowmobiles were done in 19 days. In general, the effect of large number of snowmobiles was noticeable in the $NO_x$ data only in case of low wind speed. For example, in the evening of 01.05.2017, the wind speed was 1.9 $ms^{-1}$, and the $NO_2$ concentration increased sharply to 7.3 ppb due to 21 snowmobiles passing by the station. The group of similar size was passing by in the evening of 02.05.2017, but the effect on $NO_2$ values was three times lower as the wind speed was higher (4.0 $ms^{-1}$).

*Page 18, line 382-390. The paper concludes (elsewhere) that there is high correlation between ozone at Zeppelin and Barentsburg, thus the measurements represent region ozone levels. Since ozone data from Zeppelin is available for a number of years, this regime analysis could be extended for a much longer period and thus be much more robust.*

The long-term regime and Zeppelin $O_3$ data analysis has been included in the discussion part of the paper (lines 567-611).

*Figure 8:*
*-This is not really the trajectory probability for the different regimes, but rather for the sub-periods. Having a longer (multi year) record to make these probabilities for the regimes would help. Very difficult to read the red contours.*
*- The maps are very small. There is no need to include the same label bar 9 times. Also I recommend that each map is labeled with the name of the corresponding circulation regime (applies also to figure 7).*

The figure (Figure 9 in the new manuscript) has been modified as suggested by the Reviewer 1. The long-term trajectory probability for different regimes have been included into the Discussion part of the paper (lines 566-610).

*Page 19, line 396. Why is the HYSPLIt model used for these back trajectories and not Flexpart?*

The FLEXPART dataset used in this study is a long-term global dataset, and trajectories were not run from a specific point, while the online version of HYSPLIT allows to easily set up a starting location and height of Zeppelin and Barentsburg stations and run a trajectory simulation for the extreme case studies discussed in the paper. Similarly, the combination of HYSPLIT, Lagranto and FLEXPART has been used in Trickl et al., 2020 to study stratospheric intrusion events.

*Page 22, line 474. The authors claim that "The weather regime approach is novel in the air pollution research". However, this has been used in several studies before, although not for the Svalbard region I believe.*

We would like to thank the Reviewer for this comment. The respective references have been included in the methods part (lines 165-171) in addition to another relevant article about connection of NAO and air pollution transport (Eckhardt et al., 2003; Ibrahim et al., 2021; Ménégoz et al., 2010), and line 626 has been corrected as following:

The weather regime approach is novel in Svalbard air pollution research.

**2. Reply to the Reviewer 2**

**Reply to the major comments**

*The observations itself are of interest because in the Svalbard are really sparse, so they can help to understand formation and evolution of O3 in a remote area effected by interesting and mixed effects such us depletion due to reactive halogen compounds and photochemistry due to local well defined emission sources. However, there are several shortcomings that limit a lot the results of this paper, therefore I'm really skeptical to suggest to accept this manuscript for publication on ACP, unless it will be deeply revised, for the following reasons:*
*1) Only in the Barentsburg Station are observed both O3 and NOx, that are the two species fundamental for this study, in the Adventdalen and Ny-Alesund sites are missed the measure of O3, therefore a comparison and correlations of these species among these station is misleading. To partially overcome this problem for the Ny-Alesund analysis are used the O3 measurements of the Zeppelin observatory, but this is not the solution not only because the latter is 2 km away from Ny-Alesund site, but, more seriously, it is on the top of a mountain at 474m. a.s.l., whereas Ny-Alesund site is at 23 m. a.s.l. and near the sea. Finally, since in Ny-Alesund site are missed also meteorological measurements, for the analysis on this site are used data of the top of the mountain (Zeppelin station). It would have been much more worth to install the NOx analyser in the Zeppelin station where meteorological data and O3 measurements were available then in the Ny-Alesund site.*

We agree that the lack of $NO_x$/$O_3$ instrument co-location in Adventdalen and Ny-Ålesund during the 2017 campaign is a major drawback of this study. The $NO_x$ monitoring in Ny-Ålesund is a long-term ongoing air quality project, and relocation of the instrument from the village to the Zeppelin station would introduce bias in the long-term observations in the settlement. The study in Adventdalen was the first combined air pollution and meteorological field work in Longyearbyen. The measurements there were done by the main author, and only $NO_x$ monitor was installed there due to the limited grant funding.
We would like to specify that the meteorological data from Ny-Ålesund were not missing. The data from the meteorological station operated by the Norwegian meteorological institute located 100 m away from the monitor have been used in combination with $NO_x$ data from the village (p. 5 line 120 in the manuscript). The $O_3$ and CO data from the Zeppelin station have been combined with the meteorological data from the Zeppelin station.

Regarding $NO_x$ and $O_3$ data from Ny-Ålesund, the authors have been in contact with atmospheric scientists from Ny-Ålesund research community, and, to our knowledge, the only collocated $NO_x$ and $O_3$ measurements from this settlement were performed at the Zeppelin station from February to May 1994. The results were published in Beine et al., 1996. In that study, the combination of $NO_x$ data and concentration of particles with diameter below 10 nm, atmospheric stability and wind direction was used to identify possible local pollution events. In spring 1994, the local pollution was detected at the Zeppelin station during 6.4 % of the measurement time, and the number of these events was increasing with increased insolation in May. Following this method of event detection, the concentration of particles with diameter of 10 nm routinely measured by DMPS at the Zeppelin station and threshold of > 95 percentile have been used to identify peaks in concentration of newly formed particles. Similarly to the results of Beine et al., 1996, the peak events were detected at the Zeppelin station only in the second part of the 2017 measurement campaign (from 24th of April to 13th of May). The northerly wind direction was present only during 12 out of 45 hours with peak particle concentration, however, none of these cases was characterized by increased CO concentration at the Zeppelin station. Thus, these peaks in concentration of small particles might have been related to natural rather than anthropogenic emission sources. Indeed, Heintzenberg et al., 2017 described the offset in new particle formation towards late spring and summer when biological emissions become important sources for this process. Therefore, both statistical comparison of the $O_3$ concentrations in clean and potentially

polluted air masses mentioned in the manuscript and absence of coinciding peaks in particle concentration and CO concentration indicate that the $O_3$ observations at the Zeppelin station were not significantly affected by local $NO_x$ pollution during 2017 campaign. Therefore, the lines 226-244 in the section 2.5 have been rewritten and the results about influence of local $NO_x$ pollution on the $O_3$ concentrations at the Zeppelin station have been modified accordingly (lines 288-305 in the new version of the manuscript).

*2) Lines 233-236: Since the CO measurements were stable at Zeppelin station so no sharp peaks to identify local emission were detected, this is a proof that the measurements at Ny-Alesund are not useful to understand potential impact of local pollution on O3 evolution in the mountain top.*

We agree with the Reviewer about this point, and as it is stated above, the lines 226-244 in the section 2.5 have been rewritten and the results about influence of local $NO_x$ pollution on the $O_3$ concentrations at the Zeppelin station have been modified accordingly (lines 288-305 in the new version of the manuscript).

*3) Lines 243-247: Here the Author affirm that local emissions are important because the correlation between NOx measurements at Ny-Alesund and Adventdalen sites are weak, but at lines 233-236, looking at CO data they assert that local pollution are not important for the area under investigation. This is contradictory conclusion is due to another big issue: since CO measurements are available only at Zeppelin station, the signature of local emissions in other sites (where CO were not measured), were tried to find in the correlation between NOx observation, again measurements of CO in Ny-Alesund and Adventdalen site would have been worth to make this conclusion.*

The sentences have been modified as following (lines 271-276 in the new manuscript):
Low correlation between the $NO_x$ values at the three stations indicates the importance of local emission sources and micrometeorology (wind channelling and spatial variation in atmospheric stability) rather than synoptic meteorological conditions. The background $NO_x$ concentrations observed in Svalbard in previous studies (Beine et al., 1997) using different measurement techniques are below 0.4 ppb, and thus, the natural variability in $NO_x$ values due to long-range transport to Svalbard would be undetected in the $NO_x$ datasets presented in the current study.

*4) Lines 255-257: Author here assert that synoptic transport is more important than local emission looking, now, at the correlation of O3 measurements at Zeppelin and Barentsburg site, a couple of issues: a) r = 0.69 is considered a 'strong correlation', it means r2 = 0.47, that is not that 'strong', b) again since O3 is missed in the Ny-Alesund and Adventdalen, now to decide if dominate local emission or synoptic transport the correlation od O3 between Zeppelin and Barentsburg site are used, while before (lines 233-236) were used NOx for Ny-Alesund and Adventdalen sites, obtaining contradictory results.*

a) the word "strong" has been changed to "moderate"; b) as mentioned above, the lines 226-244, 271-276 and 288-305 in the new version of the manuscript have been rewritten.

*5) Lines 295-298 and figure 4: The O3 diurnal cycle of Barentsburg site is, as expected, completely different of that of the Zeppelin station, in the first is evident the typical profile dominated by photochemistry, in the later, a typical mountain station data, with no diurnal cycle. This is what expected, but again in contrast with what reported in lines 255-257 where Author affirm that the O3 measured in these two sites showed a 'strong correlation': they have a completely different*

*dynamics, as can be expected since one is in a mountain and the other in a site at 40 m. a.s.l.*

The standard error of the mean has been included in the Figure 5 (former Figure 4) for each hour. To improve readability of the figure, separate subplots a) and b) have been made for $NO_x$ and $O_3$ data, and supporting text has been modified accordingly. As noted by the Reviewer 2, the Zeppelin station is located at the altitude of 474 m a.s.l. and mostly samples air from the free troposphere with higher $O_3$ concentration, and thus the data from this station does not exhibit similar diurnal variation as the Barentsburg station. However, the magnitude of these effects is small, and according to the t-test and the WRS-test, there is no statistically significant difference between the nighttime and daytime $O_3$ concentrations measured at the stations (Table 1).

*6) Lines 385-399: Here initially, looking at trajectory analysis Authors affirm that the O3 decrease can be explained by local depletion due to air masses rich of BrO, whereas at the end is supposed that may be due to less photochemistry due to 'lack of sunlight and O3 precursors such as NOx'. A good result of this analysis would have been if the two effects (depletion due to BrO vs photochemistry and NOx emission) were well characterized and, from observations and model analysis, quantified and compared, unfortunately here both effects are claimed, as can be guessed even without any kind of measurements and/or model analysis.*

Following recommendations of the Reviewer 1, we included investigation of possible downward transport of $O_3$ enriched air masses in the trajectory analysis and added the percentage of trajectories descending from higher altitudes (>2000 m) (Figure 9 in the new manuscript). As in previous studies of Hirdman et al. 2009, the downward transport of $O_3$-enriched air masses from higher altitudes played significant role during the 2017 campaign. The percentage of trajectory points reaching elevations above 2000 m was highest for the sub-periods III, VI and IX (27%, 33% and 24% of the total number trajectory points for each sub-period respectively). In contrast, during the sub-period VIII with the lowest $O_3$ concentration at both stations, the percentage of elevated trajectory points was minimal, only 4%. One can also see that the percentage of elevated trajectories varies for the same type of weather regime and determines importance of the downward air mass transport for the measured surface $O_3$ concentrations in different sub-periods (e.g. ScTr regime in Figure 8c) and e) and Table2). (lines 452-455).
The discussion about $NO_x$ transport and PAN decomposition modelling has been included in the Discussion part of the paper (lines 501-522).

*7) Lines 408-411: Finally, the Authors, looking O3 sondes data confirm that Zepellin data are different of that in Barentsburg because one is at the top of Mountain and the other at 40 m. a.s.l. I think that this would have been the first analysis of the paper, and not the last one after different correlation analysis where was mentioned that O3 data of those sites are strongly correlated.*

Indeed, there is a difference between the $O_3$ levels at the Zeppelin station and in Barentsburg as revealed in the radisonde data, but moderate correlation is still valid as both sites are influenced by the same long-range transport events. The analysis of long-term data from the Zeppelin station included in the Discussion part of the new version of the manuscript (lines 566-610), allows to investigate influence of different weather regimes and obtain more robust results.
As suggested by the Reviewer 2, the Figure 10 (Figure 4 in the new manuscript) and discussion about the vertical differences is moved in the beginning of the Results part, before discussion about diurnal variation and Figure 5.

*8) Lines 471-473: From the data and analysis I'm not comfortable with the conclusion that local emission of NOx may reduce O3 level by few percent in the Ny-Alensund site, since O3 there is not measured, but, again, are used for this conclusion O3 measured at the top of mountain. Here for example, since O3 where not measured would have been worth to use a Box model (such as MCM) to model O3 at Ny-Alensund, constrained by local NOx measurements.*

8) To investigate the influence of local $NO_x$ emissions on the $O_3$ concentration in Ny-Ålesund, the following part has been included in the Discussion part of the paper (lines 529-543).
To investigate the influence of local $NO_x$ emissions on the $O_3$ concentration in Ny-Ålesund, as it is required in the third hypothesis stated in the introduction of the current paper, we may use historical observations. The data from only six $O_3$ sonde launches were available for the 2017 campaign (Figure 4). However, the long-term data below 100 m from the $O_3$ sonde profiles may be used to study influence of the local $NO_x$ pollution in Ny-Ålesund on the $O_3$ concentration. These observations are suitable for this purpose because the $O_3$ sonde launching facility is located just 200 m to the south-south-west and 500 m to the south from the $NO_x$ monitor and diesel power plant, respectively. Thus, when the monitor detected $NO_x$ concentration above long-term springtime average in the launch hour, the influence of locally polluted air masses might have been observed in the lowest $O_3$ sonde data. There were in total 59 $O_3$ sonde launches, for which $NO_x$ monitor data was available in spring 2009, 2010, 2015, 2016, 2017 and 2018. The $O_3$ profile data in the lowest 100 m have been extracted for all 59 launches and grouped according to the $NO_x$ concentration detected by the monitor and wind direction in the $O_3$ sonde profiles: 1) above mean $NO_x$ concentration and northerly wind direction; 2) below or equal to mean $NO_x$ concentration and northerly wind direction. The median and mean $O_3$ values below 100 m in the group where the $NO_x$ values were above $NO_x$ mean were 11 % and 15% lower, respectively, than for the second group with northerly winds, but without elevated $NO_x$ concentration. Thus, the $O_3$ concentration in lowest 100 m downwind from the power plant in the settlement may be reduced significantly due to local $NO_x$ emissions, but the frequency of such events is unknown in absence of continuous $O_3$ measurements in the village.

*9) Lines: 484-486: From data analysis and model simulation it is hard to support these conclusions.*

The sentence has been modified as following (lines 636-367):
In contrast to $NO_x$, the concentrations of $O_3$ in Barentsburg and at the Zeppelin observatory are moderately correlated and depend on synoptic conditions that promote transport of air masses enriched or depleted in $O_3$. In other words, both these stations are regionally representative for the $O_3$ concentrations.

*A general comment and a suggestion for further observations in this area: NO2 measurements in remote area, where the concentrations are very low may have bias or instruments are below the detection limits for most of the times. There are several papers that suggest to use instrument that measure directly NO2, using CAPS, LIF or CRDS techniques (actually CAPS are commercially available now) or, at least, chemiluminescence systems that uses photolytic conversion of NO2 into NO, besides systems like those used in this work (model T200) that uses molybdenum oxide converters (Steinbacher et al., 2007; Dunlea, et al. 2007; Yang et al., 2004; Villena et al., 2012).*

We would like to thank Reviewer 2 for the suggestion of the alternative $NO_x$ measurement techniques. We agree that instruments used in this study are more suitable for urban air pollution studies, however, we decided to install the chemiluminescence analyser for the 2017 campaign in Adventdalen to make measurements comparable with data of the same type from Ny-Ålesund and Barentsburg and to take advantage on the available long-term $NO_x$ data from Ny-Ålesund that is included in the new version of the manuscript.

We will definitely consider suggested measurement techniques for background $NO_x$ observations during planning of new field campaigns in the Arctic.

**Reply to minor comment**

*Line 391: When in a time period, 67% of data are missed analysis and conclusion are very weak, so may be better not include that period in the analysis.*

The sentence has been modified as following (lines 450-451 in the new manuscript):
In the sub-period II, 67% of the data from the Zeppelin station were missing (Figure 2). The $O_3$ concentration in Barentburg was above median for this sub-period, and the trajectory data show the air masses arriving from the south-east (Figure 9b).